

# Vertical profile observations of water vapor deuterium excess in the lower troposphere

Olivia E. Salmon[1*], Lisa R. Welp[2,3], Michael Baldwin[2,3], Kristian Hajny[1], Brian H. Stirm[4], Paul B. Shepson[1,2,3+]

[1]Department of Chemistry, Purdue University, 560 Oval Dr, West Lafayette, IN, 47907, USA
[2]Department of Earth, Atmospheric, and Planetary Sciences, Purdue University, 550 Lafayette St, West Lafayette, IN, 47907, USA
[3]Purdue Climate Change Research Center, 203 S Martin Jischke Dr, West Lafayette, IN, 47907, USA
[4]School of Aviation and Transportation Technology, Purdue University, 1401 Aviation Dr, West Lafayette, IN, 47907, USA
[*]Currently at Lake Michigan Air Directors Consortium
[+]Currently at School of Marine and Atmospheric Sciences, Stony Brook University, 145 Endeavour Hall, Stony Brook, NY, 11794, USA

*Correspondence to*: Lisa R. Welp (lwelp@purdue.edu)

**Abstract.** We use $H_2O_v$ isotopic vertical profile measurements and complementary meteorological observations to examine how
boundary layer, cloud, and mixing processes influence the vertical structure of deuterium-excess (d-excess = $\delta D - 8 \times \delta^{18}O$) in the boundary layer, inversion layer, and lower free troposphere. Airborne measurements of water vapor ($H_2O_v$) stable isotopologues were conducted around two continental U.S. cities in February – March 2016. Nine research flights were designed to characterize the $\delta D$, $\delta^{18}O$, and d-excess vertical profiles extending from the surface to ≤2 km. We examine observations from three unique case study flights in detail. One case study shows $H_2O_v$ isotopologue vertical profiles that are consistent with
Rayleigh isotopic distillation theory coinciding with clear skies, dry adiabatic lapse rates within the boundary layer, and relatively constant vertical profiles of wind speed and wind direction. The two remaining case studies show that $H_2O_v$ isotopic signatures above the boundary layer are sensitive to cloud processes and complex air mass mixing patterns. These two case studies indicate anomalies in the d-excess signature relative to Rayleigh theory, such as low d-excess values at the interface of the inversion layer and the free troposphere, which is possibly indicative of cloud evaporation. We discuss possible explanations
for the observed d-excess anomalies, such as cloud evaporation, wind shear, and vertical mixing. *In situ* $H_2O_v$ stable isotope measurements, and d-excess in particular, could be useful for improving our understanding of moisture processing and transport mixing occurring between the boundary layer, inversion layer, and free troposphere.

**1 Introduction**

Water vapor ($H_2O_v$) in the lower troposphere modulates processes including cloud formation, precipitation, severe weather development, atmospheric circulation, radiative forcing, and climate feedbacks (Held and Soden, 2000; Kunkel et al., 2012; Tompkins, 2001; Trapp et al., 2007; Trenberth, 2011). Accurately representing these dynamic, mesoscale processes in models can be difficult, and efforts to improve parameterizations are on-going (Gerber et al., 2013; Park et al., 2017; Wood,
2012; Yamaguchi and Feingold, 2013). Some active areas of research include: quantifying the inversion layer entrainment flux (Wood, 2012), refining entrainment-cloud evaporation relationships (Gerber et al., 2013; Yamaguchi and Feingold, 2013), and updating cloud evaporation schemes with new cloud classes (Park et al., 2017).

Free troposphere entrainment and cloud evaporation influence the maintenance of the cloud layer, which in turn influences radiative forcing (Gerber et al., 2013; Yamaguchi and Feingold, 2013). The nature of $H_2O_v$ as a climate feedback





agent adds further complexity to our understanding of $H_2O_v$'s role in weather. Anthropogenic greenhouse gas emissions have resulted in increasing global temperatures, enhanced evaporation from soil and the oceans, and higher atmospheric concentrations of $H_2O_v$, the dominant absorber of infrared radiation (Held and Soden, 2006; Willet et al., 2007). Warmer temperatures and more humid conditions have caused a shift towards less frequent, but more intense precipitation events,

increasing the risk of both floods and droughts (Roque-Malo and Kumar, 2017; Trenberth, 2011). $H_2O_v$ also modulates production of the dominant atmospheric oxidant, the hydroxyl radical (Thompson, 1992). Thus, accurately representing $H_2O_v$ in mesoscale processes is of great importance in a warming world.

$H_2O_v$ stable isotopologue measurements are a potential tool to inform our understanding of the distribution and dynamics of $H_2O_v$ in the lower troposphere (Galewsky et al., 2016). $H_2O_v$ stable isotopologue ratios, i.e. the ratio of heavy (HDO

or $H_2^{18}O$) to light ($H_2^{16}O$) molecules, can contain information about the meteorological conditions at an air parcel's moisture source region, surface $H_2O_v$ sources, like evapotranspiration, as well as its temperature-dependent phase change history since that point (Benetti et al., 2014; Delattre et al., 2015; Lai and Ehleringer, 2011; Uemura et al., 2008; Welp et al., 2012). The $\delta$-notation indicates the sample's heavy-to-light isotope ratio reported relative to an international standard ($\delta = R_{sample}/R_{standard} - 1$), where $\delta$ is commonly multiplied by 1000 to report in units of per mil (‰).

Isotopic fractionation processes act to enrich/deplete both HDO and $H_2^{18}O$ relative to $H_2^{16}O$ in atmospheric waters, resulting in co-varying $\delta D$ and $\delta^{18}O$ signatures. Rayleigh distillation theory can be used to calculate the degree of equilibrium fractionation that occurs when condensate is removed from an air parcel as it cools, such as when it is undergoes ascent from the surface to higher altitudes. Rayleigh theory assumes that when saturation is reached, the condensate is removed immediately from the system via precipitation, thus no equilibrium occurs between the two phases. The second-order isotope parameter

deuterium excess (d-excess = $\delta D - 8 \bullet \delta^{18}O$) can be used to identify the type of fractionation occurring, equilibrium or kinetic, given that the ratio of the $\delta D$ and $\delta^{18}O$ equilibrium fractionation factors is approximately 8:1 at typical surface level temperatures (Dansgaard, 1964). The ratio of the $\delta D$ to $\delta^{18}O$ kinetic fractionation factors is typically less than 8 and decreases with decreasing relative humidity.

Observations of d-excess have been used to deduce meteorological conditions at the evaporation source, assuming it is a

conservative tracer not changed by transport and rainout processes (Benetti et al., 2014; Delattre et al., 2015; Steen-Larsen et al., 2014; Uemura et al., 2008), but there is also evidence that d-excess is not a conserved tracer of evaporative origin if other significant sources of vapor exist, especially outside of the marine environment (Gorski et al., 2015; Griffis et al., 2016; Fiorella et al., 2018, Parkes et al., 2017; Welp et al., 2012). For example, the unique d-excess signature of combustion-derived $H_2O_v$ has been used to quantify the contribution of combustion emissions to boundary layer vapor (Fiorella, et al., 2018; Gorski et al.,

2015), and several studies have demonstrated the influence of sublimation, vapor deposition, and land surface evapotranspiration on the atmospheric $H_2O_v$ d-excess signature (Casado et al., 2016; Galewsky, 2015; Griffis et al., 2016; Lai and Ehleringer, 2011; Lowenthal et al., 2016; Moore et al., 2016; Parkes et al., 2017; Schmidt et al., 2005; Samuels-Crow et al., 2014; Welp et al., 2012).

Airborne d-excess measurements may provide information about cloud processes, precipitation recycling, FT

entrainment, and more generally, the vertical structure characteristics of d-excess over different land cover and in different seasons. Measurements of d-excess have been used to estimate below-cloud precipitation evaporation (Aemisegger et al., 2015; Froehlich et al., 2008; Wang et al., 2016), and mixing between the boundary layer (BL) and free troposphere (FT) from





stationary platforms near the surface or at high-altitude mountain sites (Bailey et al., 2015; Benetti et al., 2015; 2018; Froehlich et al., 2008; Galewsky, 2015; Lowenthal et al., 2016; Samuels-Crow et al., 2014). While some high-elevation surface monitoring sites have the advantage of sampling BL and FT air over a diurnal cycle, they do not provide a complete picture of the $H_2O_v$ isotope vertical profile (VP) at a discrete point in time. Satellite measurements, which can provide discrete VP measurements, only currently provide middle troposphere δD profiles (Herman et al., 2014; Worden et al., 2012). Airborne platforms are capable of δD, $δ^{18}O$, and d-excess VP measurements at high spatiotemporal resolution, and have been conducted since the 1960s extending from the lower troposphere to the stratosphere to investigate a variety of science questions (overview in Sodemann et al. (2017)). However, relatively few airborne $H_2O_v$ isotope studies have reported d-excess measurements (Schmidt et al., 2005; Sodemann et al., 2017), due to either the study's objective or limitations of the instrumentation (Dyroff et al., 2015; Herman et al., 2014).

In this study, $H_2O_v$ stable isotope VPs were conducted in the lower troposphere during four flights around the Washington, D.C.-Baltimore, MD area in February 2016 and during five flights around the Indianapolis, IN metropolitan area in March 2016. We compare and contrast observations of the unique vertical structure of δD, $δ^{18}O$, and d-excess from three representative case study days. The case studies provide information about meteorological conditions that produce $H_2O_v$ isotopic VP profiles consistent with Rayleigh distillation theory and those where other processes must explain the observations. The case study observations reveal d-excess features unique to stratocumulus cloud evaporation and show the influence of synoptic weather patterns and urban versus rural differences on BL development. Interpretations of case study VPs are supported with observations from the remaining flight days in Washington, D.C.-Baltimore and Indianapolis areas.

## 2 Methods

### 2.1 Study sites

Flights were conducted around the Washington, D.C.-Baltimore, MD in February 2016 and around Indianapolis in March 2016. Washington, D.C.-Baltimore- is a metropolitan area of 9.8 million residents that includes the District of Columbia and encompassing parts of Maryland, Virginia, West Virginia, and Pennsylvania (U.S. Census Bureau, 2018). The Appalachian Mountains lie to the west of Washington, D.C.-Baltimore, and the Chesapeake Bay and the Atlantic Ocean lie to the east side of Washington, D.C.-Baltimore. By contrast, Indianapolis has a population of 2.0 million and is relatively isolated from other metropolitan areas by agricultural fields (U.S. Census Bureau, 2018). The closest large body of water to Indianapolis is Lake Michigan, over 200 km to the north.

### 2.2 Instrumentation

#### 2.2.1 Airborne Laboratory for Atmospheric Research (ALAR)

The Purdue Airborne Laboratory for Atmospheric Research (ALAR) is a modified twin-engine Beachcraft Duchess aircraft. The ALAR's two rear passenger seats have been removed to make room for scientific instrumentation. Ambient air at the nose of the aircraft is pulled through a forward-facing unheated 5-cm diameter PFA Teflon inlet called the "main manifold" at a flow rate of 1840 L min$^{-1}$ using a blower installed at the rear of the aircraft. Residence time in the main manifold is ≤0.1 second. Instruments sample from the main manifold with individual Swagelok "T" connections and Teflon sampling lines. The Purdue ALAR is equipped with a global positioning and internal navigation system (GPS/INS) for 50 Hz geopositional measurements and a Best Air Turbulence (BAT) probe for 50 Hz three-dimensional winds and pressure measurements (Crawford





and Dobosy, 1992). Temperature measurements are made with a microbead thermistor installed in the center pressure port of the BAT probe (Garman, 2009). Although not the focus of this study, measurements of carbon dioxide, methane, and $H_2O_v$ mole fraction were made with a Picarro G2301-m cavity ringdown spectrometer. The Picarro data frequency was 0.5 Hz and the flow rate was 850 sccm. This system provides an independent evaluation of $H_2O_v$ mole fraction measurements by the isotope analyser described in the next section. A full description of the ALAR instrumentation suite has been provided by Salmon et al. (2017).

### 2.2.2 Water vapor mixing ratio and stable isotope measurements

$H_2O_v$, δD and $δ^{18}O$ measurements (1 Hz) were made with a Los Gatos Research, Inc. (LGR) Triple Water Vapor Isotope Analyzer (TWVIA; model: 911-0034). The TWVIA was configured as a rack-mount, extended-range model, operating with an internal cell pressure of 80 Torr, and is suggested by the manufacturer for isotopic measurements over the $H_2O_v$ mole fraction range from 4,000 – 60,000 ppmv. The analyser can make measurements at $H_2O_v$ mole fractions below 4,000 ppmv, but the instrument precision worsens (discussed below). The TWVIA sampled ambient air from the main manifold at a flow rate of 500 sccm using the analyzer's internal pump. Measurements of $H_2O_v$, δD, and $δ^{18}O$ were identically lag adjusted for the sample residence time (average: 8 s) to match geopositional and meteorological measurements. Depending on the ambient air temperature, the cabin of the aircraft was heated to prevent condensation inside tubing and for the comfort of the pilot and mission scientist.

$H_2O_v$ mole fractions reported by the LGR TWVIA and the Picarro instrument were calibrated on the ground (not in flight) throughout the campaign (on 7 and 17 March 2016) using a LI-COR dewpoint generator (model: LI-610) over the $H_2O_v$ mole fraction range from 7,000 – 12,000 ppmv. This $H_2O_v$ mole fraction range corresponds to saturation vapor pressures for temperatures ranging from approximately $3^oC – 10^oC$. The LGR TWVIA (and Picarro) $H_2O_v$ mole fraction calibration curve slope, y-intercept, and $R^2$ value are 0.9845 (0.94), -280 ppmv (-200 ppmv), and 0.99978 (0.99895), respectively. Figure A1 shows that the calibrated $H_2O_v$ mole fractions from the Picarro and LGR analysers were consistent in flight. LGR $H_2O_v$ measurements are low-pass filtered relative to the Picarro measurements due to the longer LGR residence time.

The LGR TWVIA isotopic measurements were calibrated in the lab for $H_2O_v$ concentration dependence before and after the field campaign using an LGR Water Vapor Isotope Standard Source (WVISS; model: 908-0004-9003) with five standards ranging in isotope enrichment from -39.9‰ to -573.7‰ in δD and -8.7‰ to -76.2‰ in $δ^{18}O$ (Table B1). The range in the standards' δ values brackets the range of δ values measured during the campaign. The concentration dependence was characterized over the $H_2O_v$ mole fraction range from 550 ppmv – 14,000 ppmv, which corresponds to the lowest $H_2O_v$ mole fraction the WVISS could consistently emit (Appendix B) and the highest $H_2O_v$ mole fraction observed during the research flights. The TWVIA's $H_2O_v$ concentration dependence was monitored between January 2016 and June 2017, with no appreciable instrument drift observed. $H_2O_v$ concentration-dependence calibration and residual curves are provided in Fig. B1 ($δ^{18}O$) and B2 (δD), along with a discussion of the non-linear calibration curve line fitting (Appendix B). There was no need for an additional correction to normalize to the VSMOW-SLAP (Vienna Standard Mean Ocean Water – Standard Light Antarctic Precipitation) scale (discussed in Appendix B). Discussion of the instrument precision and calibration uncertainties are provided in Appendix C. Instrument precision and concentration-dependence calibration uncertainties are summed in quadrature to yield the total uncertainty in δD, $δ^{18}O$, and d-excess. Figure 1 shows the total uncertainty in δD, $δ^{18}O$, and d-excess versus $H_2O_v$. Uncertainties increase as $H_2O_v$ mole fraction decreases below 4,000 ppmv. Flight measurements of δD, $δ^{18}O$, and d-excess reported here are smoothed using a 20-second moving average which corresponds to the time required for the TWVIA-reported δ values to stabilize after a change in the sample's $H_2O_v$ mole fraction or isotopic signature (Appendix C).





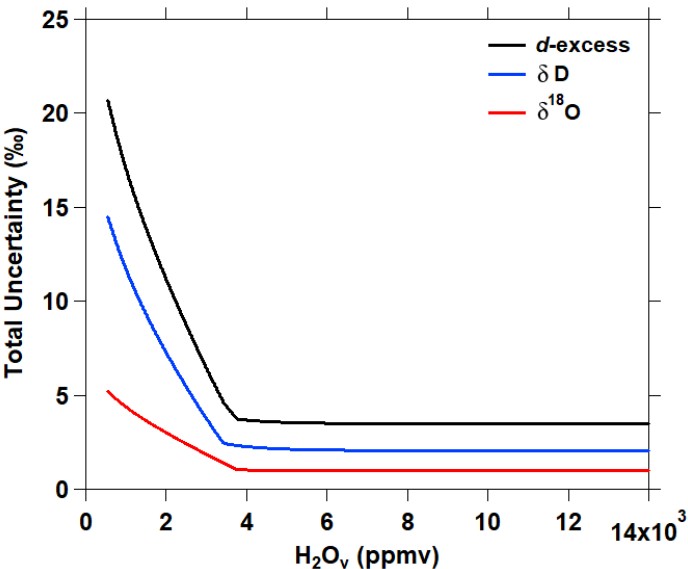

Figure 1: Total uncertainty of δD, δ¹⁸O, and d-excess over the range of $H_2O_v$ mole fractions observed in flight.

### 2.3 Flight design

5    Figure 2 shows the flight paths conducted around Washington, D.C.-Baltimore and Indianapolis. Specific flight dates and times are provided in Table 1. We focus on three particular daytime experiments conducted around Indianapolis as representative case studies (Table 1). $H_2O_v$ isotope measurements on March 6 (RAY) appear consistent with Rayleigh distillation theory, the observations on March 4 (STC) may reflect moisture processing in a stratocumulus topped-BL, and the March 18 (DBL) observations may reveal differences in urban versus rural BL development and the influence of changing synoptic

10    conditions. Conclusions about the processes influencing the isotopic features observed during the case studies are supported by measurements from the remaining Washington, D.C.- and Indianapolis flights (Table 1).



**Figure 2: Flight paths conducted around the (a) Washington, D.C.-Baltimore and (b) Indianapolis study sites for the research flight (RF) dates listed in Table 1.**



**Table 1: Flight log listing flight date, research flight (RF) and case study codes used in this manuscript, flight time (local time, LT), number of vertical profiles conducted, and the observed range of potential temperature (θ) and ambient temperature (T) during the flights.**

| Flight Date (2016) | Research Flight Code | Case Study Code (support study) | Flight Time (LT) | Vertical Profiles | θ (°C) | T (°C) |
|---|---|---|---|---|---|---|
| 12 February | RF01 | *STC*[*] | 11:45 – 17:30 | 1 | -3.0 – 6.9 | -8.9 – 9.3 |
| 17 February | RF02 | *STC*\* | 11:40 – 18:15 | 1[†] | 6.4 – 12.5 | 1.6 – 10.4 |
| 18 February | RF03 | *STC*[*] | 12:10 – 17:25 | 1 | -0.4 – 17.7 | -6.1 – 10.7 |
| 19 February | RF04 | *STC*[*] | 11:55 – 17:10 | 1 | 0.9 – 14.6 | -0.6 – 10.1 |
| 4 March | RF05 | **STC** | 13:55 – 16:30 | 5 | 3.5 – 15.4 | -2.8 – 4.2 |
| 6 March | RF06 | **RAY** | 12:55 – 15:25 | 4 | 9.6 – 21.1 | 4.4 – 11.6 |
| 7 March | RF07 | *RAY*[*] | 14:10 – 16:45 | 6 | 15.8 – 26.5 | 10.2 – 18.7 |
| 17 March | RF08 | *RAY* | 12:15 – 15:00 | 2 | 13.6 – 17.5 | 1.6 – 17.3 |
| 18 March | RF09 | **DBL** | 11:40 – 14:20 | 4[†] | 7.8 – 17.8 | 0.2 – 10.7 |

[*]The supporting research flight days share similarities with the indicated case study, but some caveats exist (Discussion 4.4).

[†]Measurements of meteorological variables are completely or partially unavailable during one of the vertical profiles due to temporary failure of winds measurement system.

Flight paths were designed to maximize the number of VPs conducted while also characterizing upwind/downwind gradients in $H_2O_v$ isotopic signature. VPs were sometimes conducted in a spiral pattern to limit the horizontal spatial coverage of

the measurements, while other VPs were conducted in a sawtooth pattern ("porpoising"; Gerber et al., 2013) between the BL and FT when the research aircraft travelled between the West Lafayette, IN, Purdue airport and the Indianapolis study site. Figure 3 shows examples of these two types of VPs conducted during the case study flights. The aircraft flew up to ~1600 m above sea level (msl) on average during the VPs. Only data collected on the descents of the VPs, when sampled air transitions from relatively dry to relatively humid, are presented here to minimize the potential influence of memory effects. However, similar

features were observed on the ascents and descents. The number of VPs (Table 1) conducted on each flight was limited by air traffic and restricted air space (which was worse for the Washington, D.C.-Baltimore study site), cloud cover, and available flight time. The research aircraft typically does not fly through clouds during experimental flights. Flights included other maneuvers, such as transects conducted upwind, intersecting, and downwind of the urban centers, the interpretation of which is beyond the scope of this paper (Fig. 3).

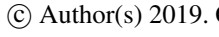



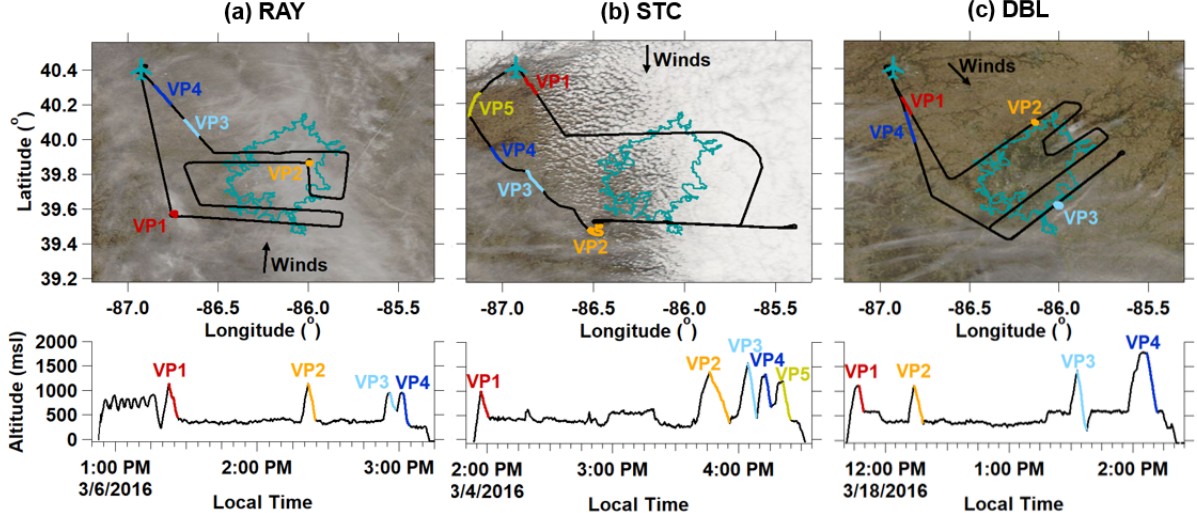

**Figure 3: The (a) RAY, (b) STC, and (c) DBL flight path and altitude time series with vertical profiles (VP) highlighted. Flight paths overlay the study site's cloud cover captured at approximately 12:30 local time by the Terra MODIS satellite** 5 **(https://worldview.earthdata.nasa.gov/). The teal outline indicates the Indianapolis city boundaries. The West Lafayette, IN, Purdue airport is indicated by the airplane marker.**

## 2.4 Atmospheric layer identification

We classify regions of the atmosphere into the boundary layer (BL), inversion layer (INV), and free troposphere (FT), 10 to compare and contrast features observed in $\delta D$, $\delta^{18}O$, and d-excess signatures during the research flights. The altitude at the base of the INV ($z_{INV}$) is defined as the lowest altitude at which the change in potential temperature ($\theta$) with altitude exceeds 0.5 K for a 10 m change in altitude ($d\theta/dz > 0.05$ K m$^{-1}$). Rates of $d\theta/dz > 0.05$ K m$^{-1}$ were commonly observed within the INV during the research flights. The altitude at the base of the FT ($z_{FT}$) is defined as the altitude above $z_{INV}$ at which $d\theta/dz$ transitions to <0.05 K m$^{-1}$. A recent evaluation of methods for determining boundary layer height from aircraft measurements indicate the 15 potential temperature gradient approach is most reliable (Dai et al., 2014). However, if layers are not definable using the $d\theta/dz >$ or $< 0.05$ K m$^{-1}$ criterion, the secondary criterion of $|d(H_2O_v)|/dz > 20$ ppmv m$^{-1}$ and $|d(H_2O_v)|/dz < 20$ ppmv m$^{-1}$ is used to define $z_{INV}$ and $z_{FT}$, respectively. These threshold values are appropriate for our wintertime, mid-latitude observations, but may not be universally appropriate in different locations or seasons. If neither criterion is met, the profiles of $\theta$, $d\theta/dz$, $H_2O_v$, and $d(H_2O_v)/dz$ are collectively considered, and $z_{INV}$ is visually defined as the point at which $H_2O_v$ and $\theta$ begin decreasing and increasing, 20 respectively. Similarly, $z_{FT}$ is visually determined as the altitude at which the rate of change of $H_2O_v$ and $\theta$ with altitude begins to decrease.

## 2.5 Isotope theory

There are three common ways the isotopic composition of atmospheric $H_2O_v$ can change: Rayleigh distillation as air is dehydrated as it cools with altitude, mixing of different air masses, and cloud formation and evaporation. We show here how 25 each of these processes is expected to change the isotopic signatures of atmospheric $H_2O_v$ as $H_2O_v$ mixing ratios change.

As air is dehydrated, for example during ascent, the heavier isotopologues are preferentially condensed first. The Rayleigh distillation model describes the effects of equilibrium fractionation on the isotopic composition of a dehydrating air





parcel (Dansgaard, 1964). Condensate that is formed as an ascending air parcel expands and cools is isotopically enriched relative to the vapor and in the open form of the Rayleigh model is assumed to be immediately removed from the system. The isotopic composition of the parcel as predicted by Rayleigh theory is given by the eq. (1).

$$R_{Ray} = R_o \left( \frac{H_2O_v}{H_2O_{v_o}} \right)^{\alpha_e - 1}$$  (1)

Here $R_o$ and $R_{Ray}$ are the heavy to light isotopologue ratios ($\frac{HDO}{H_2O}$ or $\frac{H_2{}^{18}O_v}{H_2O}$) of the parcel prior to the ascent and at any point throughout the ascent, respectively. The remaining fraction of $H_2O_v$ left in the ascending parcel relative to initial conditions is given by $\frac{H_2O_v}{H_2O_{v_o}}$. We determined the initial $R_o$ and $H_2O_{v_o}$ input values for each day from the average BL values measured along the VP descents. The temperature-dependent equilibrium fractionation factor, $\alpha_e$, for each isotopologue is calculated for the temperature corresponding to the air parcel's lifting condensation level (LCL) altitude using Horita and Wesolowski (1994) for

LCL temperatures greater than 0°C and Ellehøj et al. (2013) for LCL temperatures less than 0°C. The LCL is the height at which an air parcel would become saturated if lifted adiabatically and is often used as an estimate of cloud base height (Romps, 2017). The VP observations show that ambient temperatures vary with altitude along the vertical profiles. However, Rayleigh distillation curves calculated with $\alpha_e$ values defined by the varying ambient temperatures measured along the vertical profiles are nearly identical to Rayleigh curves calculated with a single LCL-defined $\alpha_e$ value (Figure S1).

The mixing of two air parcels (*A* and *B*) results in a heavy-to-light isotopologue ratio of an air parcel, $R_{mix}$, given by eq. (2) using $HDO_v$ and $H_2O_v$ ($H_2{}^{16}O_v$) as an example, $\frac{HDO}{H_2O}_{mix}$. In eq. (2), $R_{mix}$ is the ratio of the weighted average of the heavy isotopologue to the weighted-average of the light isotopologue. The fraction of air parcel *A*, $f_A$, and air parcel *B*, $f_B$, sum to unity. The mixture's $H_2O_v$ mole fraction is simply the weighted average of each parcels' individual $H_2O_v$ mole fraction. $H_2{}^{18}O_v$ can replace $HDO_v$ in eq. (2).

$$R_{mix} = \left( \frac{HDO_v}{H_2O_v} \right)_{mix} = \frac{f_A[HDO_v]_A + f_B[HDO_v]_B}{f_A[H_2O_v]_A + f_B[H_2O_v]_B}$$  (2)

The isotopic influence of cloud evaporation on the surrounding water vapour is complicated and depends on the mass of water in the vapour and liquid phases (Noone, 2012). Here, we compare two simplified approaches, described in detail below, but outlined here. In the first approach, we use the model from Worden et al. (2007) to describe cloud evaporation into a completely dry atmosphere. In the second approach using the model from Stewart (1975), we assume that cloud evaporation

happens in two distinct regions of the inversion layer. First, cloud liquid is formed in equilibrium with atmospheric vapour at the LCL temperature. Next, that liquid is partially evaporated in the lower portion of the INV, changing its isotopic composition. Finally, that partially-evaporated cloud droplet is moved to the upper portion of the INV where it evaporates completely.

The Worden et al. (2007) model describes the isotopic signature of an air parcel that is influenced by the evaporation of cloud droplets using a modified Rayleigh model shown in eq. (3).

$$\frac{\partial \delta}{\partial H_2O_v} = \frac{1}{H_2O_v} \left[ a_e \left( \frac{1 - f_{evap}/a_k}{1 - f_{evap}} \right) - 1 \right]$$  (3)

Here $\frac{\partial \delta}{\partial q}$ represents the change in the air parcel's δ signature with change in $H_2O_v$ concentration. The fraction of the cloud droplet that has evaporated is given by $f_{evap}$. The kinetic fractionation coefficient is given by $a_k$, and is calculated according to Merlivat and Jouzel (1979). This is a simplified model that assumes the relative humidity at the surface of the evaporating cloud droplet is 0% (Worden et al., 2007; Noone, 2012).

The isotopic signature of a cloud droplet that undergoes partial evaporation and kinetic isotope fractionation within the INV is calculated according to eqs. (4) through (6), from Stewart (1975).





$$R_{cloud} = \gamma R_{vap} + (R_{cloud,o} - \gamma R_{vap})f_{cloud}{}^{\beta} \tag{4}$$

$$\gamma = \frac{\alpha_e h}{1 - \alpha_e \left(\frac{D}{D_i}\right)^n (1-h)} \tag{5}$$

$$\beta = \frac{1 - \alpha_e \left(\frac{D}{D_i}\right)^n (1-h)}{\left(\frac{D}{D_i}\right)^n (1-h)} \tag{6}$$

Here $R_{cloud}$ is the isotopic ratio of the remaining cloud droplet, $R_{cloud,o}$ is the initial isotopic ratio of the cloud droplet, $R_{vap}$ is

the isotopic ratio of the atmospheric vapor, $f_{cloud}$ is the fraction of the cloud droplet remaining, and $h$ is relative humidity. The

ratio of the diffusivity of light water to heavy water, $\frac{D}{D_i}$, is 1.02512 for $H_2{}^{16}O$:HDO and 1.02849 for $H_2{}^{16}O$:$H_2{}^{18}O$ (Merlivat,

1978). The scaling constant, $n$, is 0.58 and determines the magnitude of kinetic isotope fractionation (Stewart, 1975).

## 3 Results

### 3.1 Rayleigh-consistent observations (RAY)

Four VPs consistent with Rayleigh distillation theory were conducted on 6 March 2016 in Indianapolis (RF06; "RAY

case study"). Observations of δD, $\delta^{18}O$, and d-excess measured along the VP descents are plotted as a function of $H_2O_v$ mole

fraction in Fig. 4a-c, respectively, along with predictions from Rayleigh distillation theory. The mixing lines in Fig. 4 show an

air parcel's isotopic signature if varying proportions of BL and FT air are mixed (Methods 2.5). For the most part, the Rayleigh-

predicated $\delta^{18}O$, δD, and d-excess values along the four VPs are consistent with the observations up to the top of the INV (Fig.

4a-c). However, d-excess observations at the interface of the INV and FT exhibit a hyperbolic shape (Fig. 4c), which is

associated with mixing between distinct air parcels (Noone, 2012). Additionally, the δD, and to a certain extent, the $\delta^{18}O$

observations deviate slightly below the Rayleigh predictions in drier portions of the INV and in the FT (Fig. 4a-b).



**Figure 4: Comparison of vertical profile δ¹⁸O (top panels), δD (middle panels), and d-excess (bottom panels) measurements to Rayleigh theory (solid) and mixing (pink) curves for three case study days RAY (RF06), STC (RF05), and DBL (RF09). Individual VP descents are indicated by the different-colored points. The average $H_2O_v$ mole fractions observed within the INV (between $z_{INV}$ and $z_{FT}$) on each days' VPs are indicated with grey shading.**



Figure 5a shows vertical profiles of potential temperature ($\theta$), $H_2O_v$, $\delta^{18}O$, $\delta D$, RH, d-excess, wind direction, wind speed, vertical wind variance (W $\sigma^2$), and ambient temperature measured along the second RAY VP (VP2). VP2 observations are presented as a representative example of RAY because VP2 was conducted approximately midway through the flight, it covers the largest vertical range relative to the remaining VPs, and it was conducted in a spiral formation to minimize the horizontal

spatial extent over which the measurements were made (Fig. 3a). VP2 measurements in the BL, from 380 m – 780 m above ground level, of $\delta^{18}O$, $\delta D$, and d-excess are relatively constant with altitude, varying by 1.2‰, 15.3‰, and 10.9‰, respectively. The ambient temperature profile approximately follows the dry adiabatic lapse rate to the top of the BL (Fig. 5a). VP2 $H_2O_v$ mole fraction decreases by 5095 ppmv in the INV between $z_{INV}$ and $z_{FT}$ before becoming relatively stable in the FT. The VP2 INV $\delta^{18}O$ and $\delta D$ values track the $H_2O_v$ profile, decreasing by 30.8‰ and 193.2‰, respectively. Observed d-excess values in the

INV initially decrease with altitude, and then increase, varying overall by 66.6‰. Just above the INV $H_2O_v$ mole fractions near a minimum, and $\delta D$ increases while $\delta^{18}O$ decreases, causing FT d-excess values to increase rapidly. Above ~1100 m in the FT, the VP2 $H_2O_v$, $\delta^{18}O$, $\delta D$, and d-excess signatures are relatively constant with altitude.

Wind speed along the RAY VP2 ranged from 4.3 m s$^{-1}$ to 10.3 m s$^{-1}$, and wind direction only varied by 60$^\circ$ from the BL to FT. Cloud top height estimated from the Terra MODIS satellite retrievals (https://worldview.earthdata.nasa.gov/) indicate that

the sparse cloud cover shown in Fig. 3a corresponds to higher altitude (>4800 m) clouds. The RAY measurements were made below 1400 m above ground level (Fig. 3a), and as a result, were likely not impacted by cloud processes from the sparse, higher altitude clouds.

Observations on 17 March 2016 (RF08) were also consistent with Rayleigh theory (Table 1). Like RF06 (RAY), skies were clear of clouds and wind speed and wind direction was relatively constant from the BL to the FT and a nearly dry adiabatic

lapse rate was present from the surface up to ~3 km on RF08 (Fig. S2). RF06 was chosen for the Rayleigh case study over RF08 because RF08 $H_2O_v$ mole fractions covered a smaller range and only two VPs were conducted that day.





Figure 5: Observations of meteorological and isotope variables along the second VP (VP2) conducted on (a) RAY and (b) STC. Measurements in the boundary layer (BL), inversion layer (INV), and free troposphere (FT) are indicated for reference. The dashed grey line in the Fig. 5b corresponds to stratocumulus cloud base ($z_{CB}$).





### 3.2 Stratocumulus-topped boundary layer observations (STC)

Figure 3b shows that the center and eastern portions of the Indianapolis study area were covered by stratocumulus clouds during part of 4 March 2016 (RF05, STC case study). Measurements on STC reveal unique d-excess features within the INV that may reflect stratocumulus cloud evaporation. Five VPs were conducted on this day. STC VP $\delta^{18}O$ observations are

relatively consistent with Rayleigh predictions (Fig. 4d), but are more negative in drier portions of the INV and FT (Fig. 4b). The grey shading in Fig. 4 is the average range of observed INV $H_2O_v$ mole fractions and does not represent the INV location for every VP on a single day. STC deviations from Rayleigh theory are more pronounced for $\delta D$ and d-excess than $\delta^{18}O$ (Fig. 4e-f). With the exception of VP1, the rest of the VPs' $\delta D$ values are more negative relative to Rayleigh in the INV and plateau in the FT (Fig. 4e). The d-excess measurements along VP2 through VP5 reveal two anomalies (Fig. 4f), (1) the slight increase in d-

excess in the middle of the inversion layer (particularly for VP2 and VP5) and the (2) d-excess minimum at the INV-FT interface ($z_{FT}$). From this minimum at $z_{FT}$, the FT d-excess signature becomes more positive with increasing altitude (as the air becomes drier), and eventually transitions to being more positive than the Rayleigh curve (Fig. 4f).

VP1 was conducted immediately after take-off from the Purdue airport. Sky conditions in the vicinity of the airport were clear, but a layer of stratocumulus clouds was observed over Indianapolis. Only one VP (VP1) was conducted before the

research aircraft encountered the cloud layer (Fig. 3b). Unlike VP2 − 5, VP1 d-excess tracks the Rayleigh line at the INV-FT interface. Slightly above $z_{FT}$, the VP1 $H_2O_v$ mole fraction began increasing and the d-excess switches to tracking the mixing line. The differences between VP1 and the other VPs are described in more detail later.

Figure 5b shows vertical profiles of $\theta$, $H_2O_v$, $\delta^{18}O$, $\delta D$, RH, d-excess, wind direction, wind speed, vertical wind variance (W $\sigma^2$), and ambient temperature measured along VP2 conducted on STC. VP2 data is presented because it was

conducted approximately mid-flight and it was conducted in a spiral formation minimizing the horizontal spatial extent over which the measurements were made. Measurements of $\delta^{18}O$, $\delta D$, and d-excess within the BL varied by 3.3‰, 27.4‰, and 19.1‰, respectively. This is approximately double the variability in $\delta D$, $\delta^{18}O$, and d-excess observed within the BL along the RAY (RF06; 6 March 2016) VP2 (Fig. 5a). Within the INV, $H_2O_v$, $\delta^{18}O$, and $\delta D$ values decrease by 1930 ppmv, 17.6‰, and 159.7‰, respectively from BL values. Unlike RAY, d-excess first increases with altitude within the lower INV before

decreasing to a minimum at $z_{FT}$. Similar to RAY, d-excess steadily increases in the FT on STC as $H_2O_v$ mole fractions decrease.

One difference between STC and RAY is the presence of a stratocumulus cloud layer for STC (Fig. 3). Figure 5b shows that STC VP2 air becomes nearly saturated at 788 m ($z_{CB}$ for "cloud base"). The ambient temperature lapse rate is 8.8 K km$^{-1}$ (close to the dry adiabatic lapse rate of 9.8 K km$^{-1}$) near the surface until an altitude of $z_{CB}$, where the lapse rate transitions to 2.8 K km$^{-1}$. These observations could indicate a stratocumulus cloud layer, which sits directly below the INV, and sustains the

temperature inversion via radiative cooling (Wood, 2012). Indeed, $\theta$ decreases sharply at $z_{INV}$ (Fig. 5b). Indications of a stratocumulus cloud layer were apparent on VP1 and VP2, but a clear change in lapse rate below the INV was not observed on the remaining STC VPs, indicating the air was not saturated below the INV during the later portion of the flight.

Figure 3b shows that stratocumulus clouds covered most of the study area at approximately 12:30 local time on STC. The cloud cover map in Fig. 3b is provided to show the cloud type and extent during the afternoon of STC, however, it does not

necessarily represent the cloud cover conditions throughout the 2.5 h flight. Figure S3 is a GIF of satellite cloud cover images captured during the afternoon of STC that shows the cloud cover evolution over the course of the flight. Thick cloud cover was sustained into the middle of the flight, particularly over the city of Indianapolis. As a result, VPs were not conducted within the city limits of Indianapolis. Toward the latter half of the flight the cloud cover transitioned to scattered. Four additional VPs were conducted at the end of the flight west of Indianapolis (Fig. 3b).




In contrast to RAY (Fig. 5a), STC wind speed is highly variable from the BL to the FT (Fig. 5b). Wind speed values ranged from a minimum of 0.4 m s$^{-1}$ in the BL to a maximum in the FT of 9.8 m s$^{-1}$. A distinctive wind shear is obvious at $z_{FT}$. Highly variable wind direction in the BL is a result of low BL wind speeds.

### 3.3 Observations of a developing boundary layer (DBL)

5      The third case study measurements were conducted on 18 March 2016 in Indianapolis ("DBL" for developing boundary layer, RF09). Measurements on this day reveal considerable spatiotemporal variability in the vertical structure of the observed meteorological and isotopic variables. The boundary layer height increased over the course of the flight and may reflect a combination of a residual layer from the previous day, urban vs. rural differences in BL development, and the effects of a frontal pattern moving across the Indianapolis study area. Figure 3c shows that Indianapolis was cloud free at approximately 12:30 LT

10    and clear skies continued throughout the flight. Scattered clouds developed over Indianapolis late in the afternoon after the research flight (17:00 local time; Fig. S4). Within the BL, wind direction and wind speed were relatively constant. Wind shearing is apparent at the $z_{FT}$ (Fig. 6). Wind speeds increase from ~5 m s$^{-1}$ to ~18 m s$^{-1}$ between the BL and FT, which is a larger gradient than was observed on STC. Wind speed stabilizes within the FT (Fig. 6). Wind direction varies only by ~30 $^{o}$ during each of the four VPs.

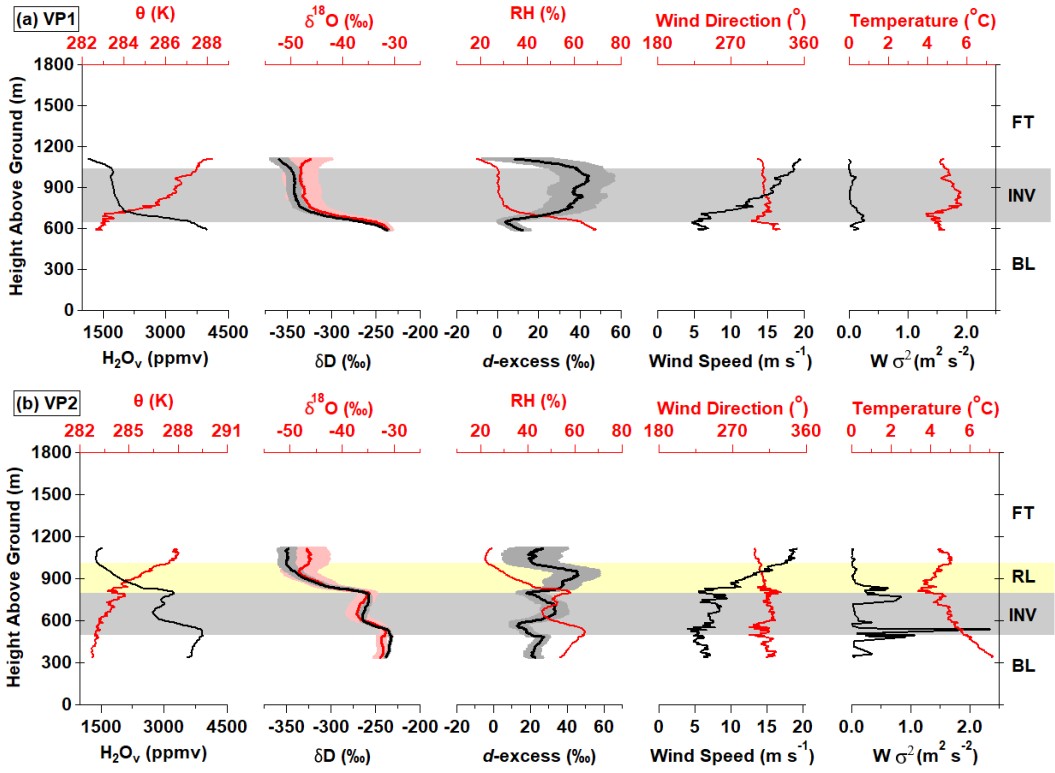





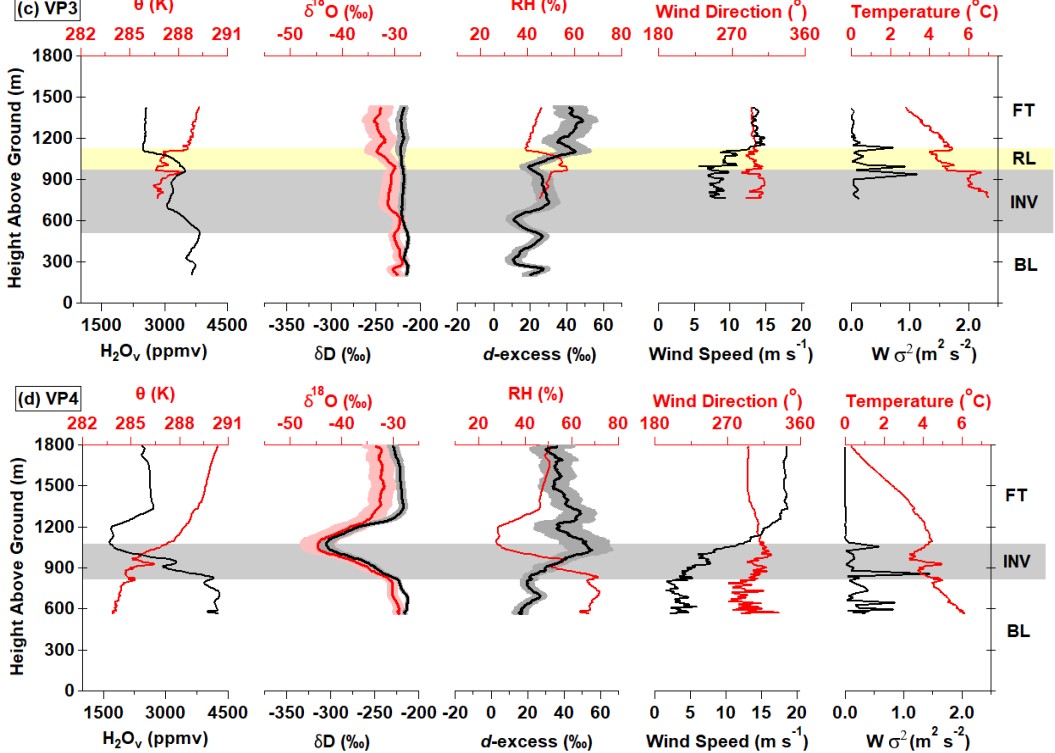

**Figure 6: DBL vertical profile (VP) measurements from RF09 in the boundary layer (BL), inversion layer (INV), previous day's residual layer (RL), and free troposphere (FT). Observations corresponding to VP1-4 are shown in (a-d), respectively.**

Differences in the vertical structure of $\delta^{18}O$, $\delta D$, and d-excess along the four DBL VPs are shown in Figure 6. There appears to be two distinct atmospheric layers separating the BL and FT in VP2 (Fig. 6b). The layer directly below the FT in VP2 is the residual layer (RL) from the previous day's boundary layer (Fig. 6b). Indications for the presence of a RL are discussed in Section 4.3. Both the RL and the INV (directly below the RL) show characteristic decreases in $H_2O_v$, $\delta^{18}O$, and $\delta D$ values. The presence of multiple layers is supported by the increase in the variance of the vertical wind speed (W $\sigma^2$), indicating wind shear, at the interface of atmospheric layers (Fig. 6b-d). The temporal factors influencing the presence of the RL are discussed below.

The $\delta^{18}O$ and $\delta D$ values were relatively constant with altitude along VP3 (Figure 6c), however the vertical structure of $H_2O_v$ and d-excess is similar in shape to VP2 (Fig. 6b). Similarities in $H_2O_v$ and d-excess features between VP2 and VP3 give indications that a RL and INV were present during VP3 as well as VP2. The winds measurement system on the aircraft had a temporary failure halfway through the VP3 descent, but available measurements show an increase in wind speed, as well as an increase in vertical wind variance (W $\sigma^2$) and small temperature inversions at the base and top of the INV (Fig. 6c). The VP4 vertical structure of $\delta^{18}O$ and $\delta D$ near the surface and higher in the FT (>1400 m) are constant, and are of similar enrichment to VP3 $\delta$ values. However, VP4 $\delta^{18}O$ and $\delta D$ values decrease to a minimum within the INV before increasing and then plateauing in the FT.





Despite differences in δ value features along the four DBL VPs, the relationship between d-excess and $H_2O_v$ mole fraction appears relatively consistent throughout the day (Fig. 4i). A positive d-excess anomaly relative to Rayleigh is present at the INV-FT interface, in contrast to the negative d-excess anomaly in STC (Fig. 4i). Inspection of the $\delta^{18}O$ (Fig. 4g) and δD values (Fig. 4h) show that δ values along VP1 and VP2 are consistent with Rayleigh predictions in the BL and in the more humid

portion of the INV. As $H_2O_v$ mole fractions decrease in the INV, the $\delta^{18}O$ and δD signatures observed on VP1 and VP2 transition to more negative values than the Rayleigh prediction, but approximately retain their original slope, suggesting agreement with the Rayleigh model if more isotopically-depleted initial conditions were considered. DBL VP3 $\delta^{18}O$ and δD values are enriched relative to VP1 and VP2, and are relatively constant between the BL and FT (recall the grey shading in Fig. 4 represents the average range of INV $H_2O_v$ mole fraction). Despite VP3 extending into the FT, $H_2O_v$ mole fractions only decreased to ~2500

ppmv, whereas $H_2O_v$ mole fractions of 1700 ppmv and less were observed in the FT of VP1, VP2, and VP4 (Fig. 4g-i). VP4 $\delta^{18}O$ and δD are similar to VP1 and VP2 in the BL through to the lower FT. However, the trend in the VP4 $H_2O_v$ mole fraction reverses in the FT, at approximately 1370 m, and begins increasing. VP4 FT $\delta^{18}O$ and δD values instead appear to track a mixing line with the VP3 δ values.

**3.4 General observations of $H_2O_v$ isotopologues in the lower troposphere**

The case study days presented above were chosen for their distinct isotopic features and because several VPs were conducted each day. However, they only represent 30% of the research flight days (Table 1). Figure 7a-c shows the Washington, D.C.-Baltimore and Indianapolis VP absolute d-excess observations in the FT, INV, and BL, respectively. Figures 7d-f show the measured d-excess relative to Rayleigh distillation theory, i.e. Rayleigh-predicted d-excess has been subtracted from the

observations. Overall, BL d-excess observations at the Indianapolis and Washington, D.C.-Baltimore study sites are relatively consistent with Rayleigh theory when using observations on each day as the initial conditions (Fig. 7c). The greatest departures from Rayleigh theory were most commonly observed in the INV (Fig. 7b). Observations of d-excess in the INV at the Washington, D.C.-Baltimore study site in particular were significantly more negative than Rayleigh predictions, by up to -80 ‰ (Fig. 7b). Very low $H_2O_v$ mole fractions were commonly observed in the FT, as were large, positive d-excess values. Large

positive d-excess values are predicted by Rayleigh distillation theory at low $H_2O_v$ mole fractions. However, observed FT d-excess values are even more positive than the Rayleigh predictions in Fig. 7a at the lowest $H_2O_v$ mole fractions.

The 7 March 2016 flight day in Indianapolis (RF07) stands out as an unusual set of conditions because the lower troposphere increased in $H_2O_v$ mole fraction with altitude (Fig. 7a-c; gold trace). A warm, southerly front moved into the Indianapolis study area on this day, and rain preceded the flight observations. The relatively high $H_2O_v$ mole fractions in the

INV and FT likely reflect residual humidity from the storm. Overall, the RF07 VP observations do not exhibit distinctive isotopic features, and d-excess generally varies around Rayleigh predictions for the BL, INV, and FT (Fig. 7).





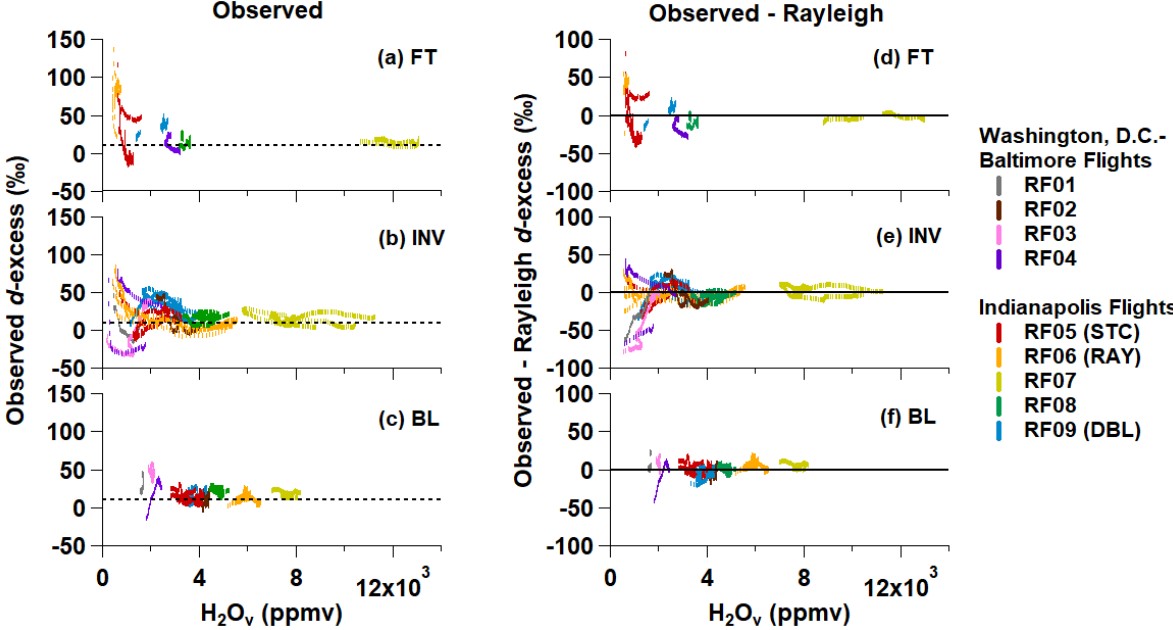

**Figure 7: Rayleigh-subtracted d-excess observations in the (a) free troposphere (FT), (b) inversion (INV), and (c) boundary layer (BL) during all the VP descents. Colors indicate each research flight. BL, INV, and FT observations are defined by the rate of change of atmospheric variables, and not defined by $H_2O_v$ mole fraction.**

Most measurements of $H_2O_v$ mole fraction in the FT were below 4000 ppmv; the corresponding FT measurements of $\delta^{18}O$, $\delta D$, and d-excess are plotted as a function of $H_2O_v$ in Fig. S5. While each Washington, D.C.-Baltimore VP (Table 1) extended into the FT, $H_2O_v$ mole fractions were often lower than the humidity range over which the TWVIA was calibrated (lower limit: 550 ppmv; Appendix B). Contributing to the scarcity of Washington, D.C.-Baltimore VP measurements is the fact that only one VP was conducted per flight, mainly as a result of congested and restricted air spaces near the capitol. Values of $\delta D$ range from -30 to -60‰, $\delta^{18}O$ ranges from -200 to -400‰, and d-excess can be close to 10‰, but also increase up to 100‰ below 1000 ppmv.

## 4 Discussion

### 4.1 Rayleigh-Consistent Conditions

Previous observations have shown that Rayleigh distillation theory successfully explains most of the variability in $H_2O_v$ isotopic composition observed in the BL (Lee et al., 2006). RAY is the simplest case study day where δ values and d-excess observations generally track the Rayleigh predictions in Fig. 4a-c. However, positive deviations from Rayleigh d-excess exist in the FT, and we hypothesize that dry FT air parcels carrying large, positive d-excess values can mix downward into more humid air parcels of smaller d-excess values near the top of the INV (Sodemann et al., 2017). As the $H_2O_v$ mole fraction approaches zero, Rayleigh-predicted d-excess approaches 7000‰ (Bony et al., 2008). Thus, lower-altitude FT air masses can carry a more positive d-excess signature for a given $H_2O_v$ mole fraction due to vertical mixing, than is predicted by Rayleigh theory (Fig. S5c). The d-excess signature in the FT more closely follows the FT-BL mixing line (Fig. 4c), which supports this hypothesis. Our results are consistent with Dyroff et al. (2015) who report lower troposphere $\delta D$ observations over the Atlantic Ocean, and





explain the vertical structure of δD at lower altitudes using Rayleigh theory, while higher altitude observations indicate mixing scenarios dictate the δD profile.

From our measurements, we can comment on what types of atmospheric conditions are likely to produce profiles consistent with Rayleigh theory, i.e. equilibrium fractionation and no vapor-condensate (cloud droplet) equilibration. The RAY

meteorological and isotopic observations are very similar to those made on 17 March 2018 (RF08; Table 1). These two flight days' observations provide possible criteria for when it is appropriate to use Rayleigh theory to predict isotopic signature throughout the BL and INV.  These criteria include that the study area is free of clouds/precipitation (supported by the presence of a dry adiabatic lapse rate throughout the BL) and wind speed and direction is relatively constant throughout the BL, INV, and FT. The STC and DBL case studies, which do not follow Rayleigh theory, violate one or more these criteria.

**4.2 Stratocumulus Cloud Evaporation**

Cloud processes may cause the d-excess anomalies from Rayleigh theory in the STC case study day. When evaluating the effect that cloud evaporation could have on vertical profiles of water vapour isotopic variability, we must consider the altitudes at which clouds form and evaporate.  In the case of evaporation, this may not occur in a single layer, but throughout a range of altitudes or atmospheric layers.  Stratocumulus cloud tops are typically present directly below $z_{INV}$ (Wood, 2012).  The

top of the INV ($z_{FT}$) is approximately the upper limit of BL mixing (Wood, 2012).  Lofting of cloud droplets into the INV would cause droplet evaporation, as the INV was under-saturated (Fig 5b).  In the cloud evaporation scenarios discussed below, it is assumed that the cloud droplets form at the top of the BL, and the droplets undergo evaporation within the INV.

Figure 8a-b shows $H_2O_v$ δD and d-excess predicted for several cloud evaporation scenarios. We choose to not show $\delta^{18}O$ under these cloud droplet evaporation scenarios because the results are similar to those for δD.  Scenario 1 in Figure 8a

shows that the δD isotopic signature of atmospheric vapor that is influenced by 35% cloud droplet evaporation air (eq. 3 in Section 2.5) tracks the STC observations for VP3 and VP4 within the INV.  While cloud evaporation scenario 1 tracks with observed d-excess in more humid portions of the INV (Figure 8b), it predicts larger positive d-excess in dryer portions of the INV (i.e. <2500 ppmv). This is the opposite of what is observed in d-excess at the upper (drier) part of the INV for VP2 through VP5. Scenario 1 does describe the steep slope in δD (and $\delta^{18}O$) values in the INV but does not explain the minimum in d-excess

(Figure 8a).



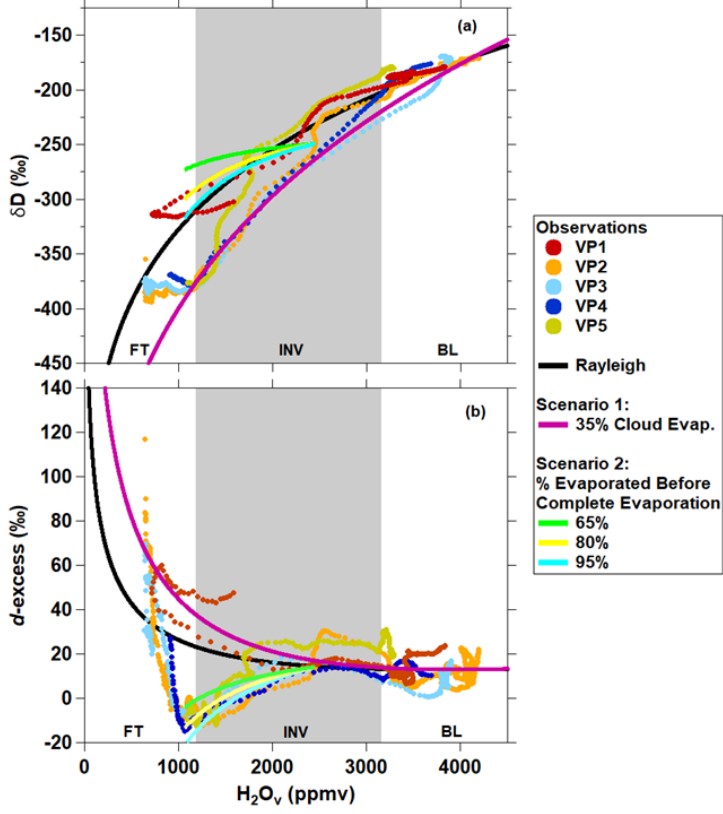

**Figure 8: STC vertical profile (VP) (a) δD and (b) d-excesss observations plotted with different cloud droplet evaporation scenarios. The two scenarios consider partial cloud droplet evaporation (Scenario 1) or the complete evaporation of previously dehydrated cloud droplets followed by mixing (Scenario 2).**

Cloud evaporation scenario 2 considers the effect of complete evaporation of a previously dehydrated cloud droplet (from eqs. 4-6 in Section 2.5). Figure 9 shows the d-excess signature of an evaporating cloud droplet that was formed at the top of the BL (consistent with stratocumulus cloud formation) evaporating into air that has an isotopic composition consistent with that observed at the top of the INV on STC. Figure 9 shows the d-excess signature of cloud droplets as they near complete

10 evaporation, highlighting 65%, 80%, and 95% evaporation values. Figure 8 shows the effect of complete evaporation of these semi-dehydrated (65%, 80%, and 95%) cloud droplets at the top of the INV, followed by subsequent mixing with INV air (eq. 2 in Section 2.5). As can be seen from Figure 8b, scenario 2 describes the minimum d-excess signature at the top of INV. However, Figure 8a shows that scenario 2 does not agree with VP2-4 δD observations within the INV. It is possible that partial evaporation (scenario 1) occurs in the lower half of the INV, followed by complete evaporation of previously dehydrated cloud

15 droplets at the top of the INV (scenario 2). We believe the minimum d-excess anomaly at $z_{FT}$ for VP2-5 may result from complete evaporation of a partially dehydrated cloud droplet (Fig. 8b). We find discussion of these potential d-excess anomalies in the literature in reference to raindrops evaporating below the cloud layer, but we believe the same isotopic fractionation would occur as liquid cloud droplets evaporate in unsaturated environments, such as at the top of the INV (Aemisegger et al., 2015; Gat, 1996; Sodemann et al., 2017).



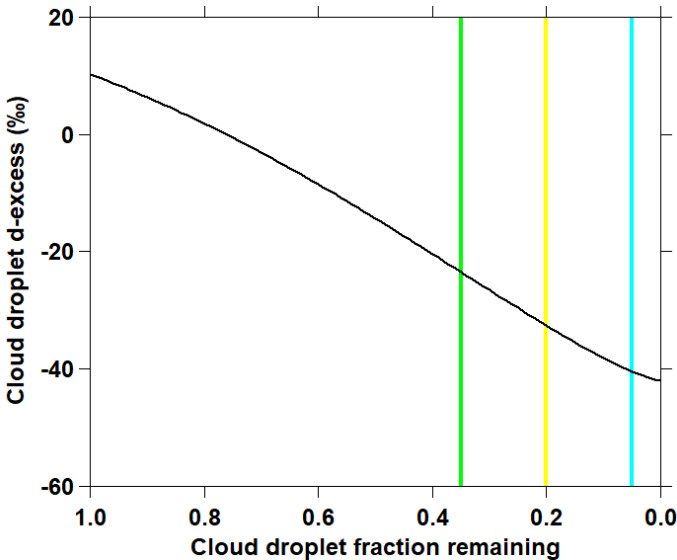

**Figure 9. Cloud droplet (condensate) d-excess as a function of the fractional amount of the droplet lost to evaporation (calculated using eqs. 4-6). As cloud droplets dehydrate, d-excess values can become extremely negative. The d-excess signature of cloud droplets that have evaporated by 65%, 80%, and 95% are highlighted. The impact on d-excess of complete evaporation of these previously dehydrated cloud droplets, followed by mixing, is shown in Figure 8.**

The slight easterly movement of the cloud layer as shown in Fig. S3 could be interpreted as the stratocumulus cloud layer being advected out of the Indianapolis study area by westerly winds rather than evaporating away. However, Fig. 5b shows that wind speeds at the cloud layer were relatively low (<6 m

s$^{-1}$), and the clouds in Figure S3 transition from opaque to semi-transparent over the flight period. This information taken together suggests there was evaporation of the stratocumulus cloud layer during the flight.

We believe our observations provide evidence that the process of cloud evaporation may be spread throughout the INV layer with the beginning and end of evaporation separated in space. Within the INV, turbulent updrafts and downdrafts are warmer and cooler than the surrounding air, respectively (Betts, 1985). Updrafts potentially carrying partially-evaporated cloud droplets to the top of the INV may facilitate complete evaporation of the droplets due to the low RH and possibly wind shear-promoted mechanical turbulence at $z_{FT}$ (Fig. 5b). Due to higher RH values within the middle of the INV relative to $z_{FT}$, it is possible cloud droplet evaporation did not proceed to completion at these INV altitudes. Partial evaporation would impart a positive d-excess signal on atmospheric vapor and act to increase RH (Aemsigger et al., 2015; Sodemann et al., 2017), both of which exhibit a local maximum at ~985 m agl (Fig. 5b).

Other studies have reported negative water vapor d-excess and hypothesize that these are from kinetic fractionation of vapor during deposition on ice crystals or snow (i.e. in ice supersaturated conditions) (Casado et al., 2016; Galewsky, 2015; Lowenthal et al., 2016; Moore et al., 2016; Samuels-Crow et al., 2014; Schmidt et al., 2005). It is unlikely that ice supersaturation is responsible for the minimum in vapor d-excess observed at $z_{FT}$ on STC because temperatures were greater than 0°C (Table 1). However, as an example, Figure D1 shows the theoretical d-excess values of STC vapor under ice supersaturated conditions (Appendix D). We reiterate that ice-supersaturation is an unlikely explanation for the STC $z_{FT}$ d-excess minimum because flight altitudes were less than 2 km, and ice (cirrus) clouds are typically present at ~6 km. It is unlikely that ice hydrometeors falling from higher altitudes could be sustained at the top of the inversion and contribute to the low d-excess signal observed on VP2–VP5 through vapor deposition given the >0°C temperatures. While vapor deposition in ice supersaturated



conditions might not be relevant for the STC flight, low d-excess (relative to Rayleigh at RH = 100 %) was sometimes observed in the INV layers during the Washington, D.C.-Baltimore flights (Fig. 7b). Ambient temperatures observed in flight in Washington, D.C.-Baltimore were sometimes less than 0$^{o}$C (Table 1), thus vapor deposition on ice crystals would be more likely for those scenarios.

### 4.3 Developing Boundary Layer

Unlike RAY and STC, the vertical structures of $H_2O_v$, δD, δ$^{18}$O, and d-excess observed in DBL are highly variable in both space and time (Fig. 6). Despite variability in the vertical structure, a fairly consistent d-excess – $H_2O_v$ relationship is maintained throughout the flight (Fig. 4i). Here we attempt to interpret and explain the heterogeneity in the DBL VPs.

A unique feature of DBL is the presence of a residual layer (RL), which was incorporated into the BL throughout the duration of the flight. This residual layer hypothesis is supported by Fig. S2, which shows that on the day prior to DBL, 17 March 2016, Rapid Refresh Model ambient temperature profiles follow a nearly dry adiabatic lapse rate all the way to an altitude of 3 km. This was a relatively warm, turbulent day and $H_2O_v$ isotopologue observations were consistent with Rayleigh theory (Table 1, RF08). A cold front moved into the Indianapolis study area on DBL (18 March 2016). The ambient temperature profile

on DBL shows the previous day's residual layer persisted into the early afternoon, between approximately 1 - 3 km (Fig. S2), before being incorporated into the BL. The distinction between the RL and BL blurs as surface heating progresses throughout the day and the RL is incorporated into the BL. It would also be expected that the downwind edge of the Indianapolis city boundaries (winds were from the northwest; Fig. 3c) would have more well mixed BL due to stronger turbulent mixing from the urban heat island and increased surface roughness (Grimmond et al., 2010; Stull, 1988). Support for this is given by DBL VP3

observations (Fig. 6c), which reveal a considerably more homogenous structure in δD and δ$^{18}$O relative to VP1 and VP2. The ambient temperatures measured along VP3 in the FT and RL are warmer relative to ambient temperatures along the other three VP's (Fig. 6), indicating the influence of the urban heat island. Although the VP3 $H_2O_v$, δD, and δ$^{18}$O values are relatively more homogenous in the vertical dimension, the d-excess signatures still maintain indications of the RL, as the vertical structure of d-excess is similar to VP2 d-excess observations. This is an example of how d-excess can provide more clues about atmospheric

circulation than can δD or δ$^{18}$O alone.

A shortwave trough in the mid-troposphere (3-5 km) carried moist air into the Indianapolis study air in the late afternoon on this day and likely influenced FT measurements (Figure S6). The dewpoint profile in Fig. S2 shows this relatively moist mid-tropospheric air descending over the course of the afternoon, reaching flight altitudes by the time VP4 was conducted. VP4 was conducted over a rural area north of Indianapolis at the end of the flight and a RL is not obvious (Fig. 6d). However,

unlike VP1-3, a sharp decrease in δD and δ$^{18}$O was observed at $z_{FT}$ on VP4 before increasing with altitude until reaching δ values observed in the VP4 BL and throughout VP3. The DBL case study shows the δ$^{18}$O, δD and d-excess measurements can be used as tracers to track the development of different atmospheric structures and circulations, including residual layers, urban heat island impacts, and passing fronts.

### 4.4 Features of the lower troposphere d-excess vertical profile

The data presented here are some of the few lower troposphere water vapour isotope observations published, so we look for common patterns that can be used to predict values in other areas. Nearly all BL observations of d-excess at the Indianapolis and Washington, D.C.-Baltimore study sites agree well with Rayleigh theory (Fig. 7c). This is consistent with previous



observations showing that Rayleigh distillation theory successfully explains most of the variability in $H_2O_v$ isotopic composition observed in the BL (Lee et al., 2006).

The FT is generally drier than the BL (Fig. 7a), and FT d-excess values observed at very low $H_2O_v$ mole fractions are often more positive than predicted by Rayleigh theory (Fig. 7a). This has been explained by very dry, depleted FT air masses,

which carry large positive d-excess signatures, mixing downward towards flight-level altitudes (Bony et al., 2008; Sodemann et al., 2017). These FT air masses likely would have originated from another source region and possibly underwent multiple condensation cycles to achieve such isotopic depletion prior to mixing with more humid air across the INV. Thus, we do not necessarily expect FT air to have a d-excess signature consistent with Rayleigh theory of an ascending BL air parcel (Dyroff et al., 2015).

We observed large departures from Rayleigh theory in the INV. Figures 7b and 7e show that d-excess observations in the Washington, D.C.-Baltimore INV can deviate negatively by ~80‰ relative to Rayleigh predictions. We believe the minimum in STC d-excess at $z_{FT}$ is a result of stratocumulus cloud evaporation. Partly cloudy or overcast conditions were also present over the Washington, D.C.-Baltimore study site on all four Washington, D.C.-Baltimore flights (https://worldview.earthdata.nasa.gov/; https://www.ncdc.noaa.gov). It is possible that the very negative Washington, D.C.-

Baltimore INV d-excess measurements were a result of complete evaporation of semi-evaporated cloud droplets within the inversion. We note that the most negative Washington, D.C.-Baltimore d-excess values correspond to the driest INV observations (Fig. 7b), where we would expect cloud top evaporation resulting from free troposphere entrainment to be the most prevalent.

## 5 Conclusions

The aim of this study is to observe and interpret the vertical structure of $H_2O_v$ stable isotopic composition, specifically d-excess, in the continental lower troposphere. Current literature regarding d-excess observations is heavily focused on ocean evaporation at coastal or island surface sites (Benetti et al., 2014; 2015; 2018; Delattre et al., 2015; Steen-Larsen et al., 2014; Uemura et al., 2008). Few reported observations of d-excess in the INV and FT exist (Galewsky et al., 2015; Lowenthal et al., 2016; Samuels-Crow et al., 2014; Schmidt et al., 2005; Sodemann et al., 2017), and, to our knowledge, only one study has used

airborne measurements to provide high vertical resolution snapshots of the lower troposphere d-excess profile at discrete time points (Sodemann et al., 2017). Our stable $H_2O_v$ isotope measurements over two continental sites is a starting point in filling the field's gap in understanding variability in the lower troposphere d-excess profile, and what it reveals about lower troposphere moisture processing on relatively small regional scales.

Our observations reaffirm the dominant role that Rayleigh distillation processes have on water vapour isotopic

variability, especially in the boundary layer. This process can predict vertical profiles up through the atmospheric inversion layer in clear-sky conditions with reasonably constant wind conditions with height. These new results highlight the potential for water vapour isotope ratios, especially d-excess to diagnose complex processes across the atmospheric inversion layer including cloud condensation, evaporation, and mixing or entrainment of free tropospheric air into the boundary layer. These types of measurements may become increasingly valuable as we seek to understand the physical processes that sustain cloud layers and

spatial-temporally variable boundary layer mixing depths.

Our interpretation of the d-excess VPs could be further evaluated by isotope-enabled circulation and weather models (Aemisegger et al., 2015; Pfahl et al., 2012; Schmidt et al., 2005). However, the simulation of convective BL processes with isotope-enabled models is complex (Benetti et al., 2018). The measurements reported here could help further develop current



and forthcoming isotope enabled models, particularly for simulating wintertime, continental lower troposphere processes or stratocumulus evaporation. Our observations of the d-excess profile in a stratocumulus cloud-topped BL and the d-excess observations reported by Sodemann et al. (2017) near marine cumulus clouds seem to indicate cloud-class specific RH and d-excess relationships. Future studies could interrogate the sensitivity of the d-excess signature to different classes of clouds and their associated unique cloud processes.

**Data availability**

Geospositional, meteorological, greenhouse gas, and water vapor isotope measurements are available for the Washington, D.C.-Baltimore and Indianapolis flight days (Table 1) are available by request and are archived in the stable water vapour isotope database: https://vapor-isotope.yale.edu/. The authors request that they be notified if the data is to be used in publication.

**Appendices**

**Appendix A. Comparison of LGR and Picarro $H_2O_v$ mole fraction**

Figure A1 compares calibrated $H_2O_v$ mole fractions from the Los Gatos Research (LGR) Triple Water Vapor Isotope Analyzer (TWVIA) and the Picarro cavity ringdown spectrometer during the entire flight conducted on 6 March 2016 (RAY).

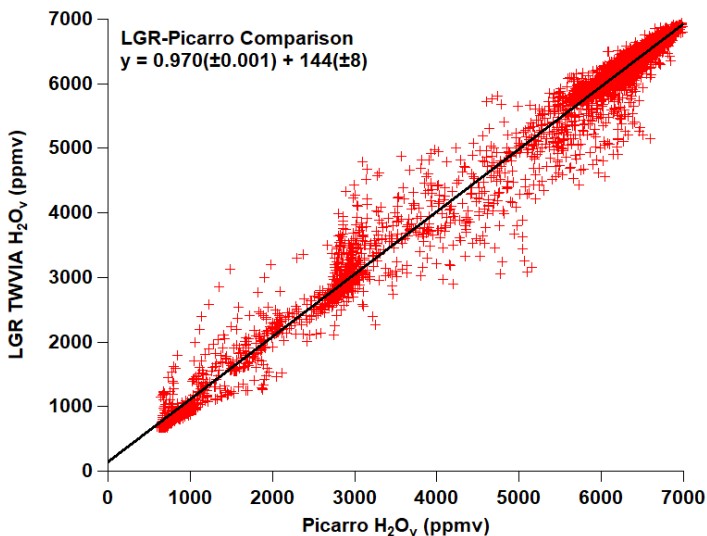

**Figure A1: Comparison of calibrated LGR and Picarro $H_2O_v$ mole fraction measurements from the entire research flight conducted on 6 March 2016 (RAY).**

**Appendix B. Water vapor concentration-dependence calibration**

A Los Gatos Research (LGR) Water Vapor Isotope Standard Source (WVISS; model: 908-0004-9003) equipped with a secondary dry air mixing chamber for extended range operation was used to characterize the LGR Triple Water Vapor Isotope Analyzer's (TWVIA; model: 911-0034) non-linear response to water vapor ($H_2O_v$) concentration (Rambo et al., 2011). The WVISS samples liquid water with a known isotopic composition from a reservoir. The standard sample is then nebulized using zero (dry) air into a heated chamber (75°C), where it evaporates completely and is further diluted with zero (dry) air with



programmable flow rates to output a range of $H_2O_v$ fractions with the same isotopic signature as the liquid standard. Different combinations of nebulizer sizes (flow rates) and standard versus extended range operation were required to span a large range of $H_2O_v$ values. The TWVIA's $H_2O_v$ dependence (while operating in extended range mode, ~80 Torr) was evaluated over the range from 550 ppmv – 14,000 ppmv, consistent with range of $H_2O_v$ mole fractions observed during the research flights (Table 1). Free

troposphere $H_2O_v$ mole fractions were sometimes less than 550 ppmv, but the lowest $H_2O_v$ mole fraction the WVISS can emit is 500 ppmv and we found stable flows of $H_2O_v$ mole fractions lower than 550 ppmv were difficult to achieve with the WVISS. We opt not to report δD and $δ^{18}O$ values for instances when $H_2O_v$ mole fraction is less than 550 ppmv. The δD and $δ^{18}O$ values of the $H_2O_v$ isotope standards, which bracket the ranges observed during the research flights (Table 1), are listed in Table B1. The WVISS was programmed to sample each $H_2O_v$ mole fraction for ≥20 min. The δD and $δ^{18}O$ $H_2O_v$ dependence calibration curves

were constructed from the average δD and $δ^{18}O$ values reported during the last 200 s of each calibration period in order to remove any influence of transition instability caused by water moving onto and off of the walls of the system during the calibration $H_2O_v$ step changes. The $δ^{18}O$ and δD $H_2O_v$ dependence curves shown in Fig. B1 and B2, respectively, were fit using the locally weighted polynomial regression "locpoly" function from R's "locfit" package (Bailey et al., 2015). A 100 ppmv sliding window was used for the local polynomial regression fitting over the range from 550 ppmv – 14,000 ppmv $H_2O_v$.

**Table B1: Calibration standards**

| Standard* | δD (‰) | $δ^{18}O$ (‰) | d-excess (‰) |
|---|---|---|---|
| Purdue tap water | -39.9 | -8.7 | 29.7 |
| Boulder tap water | -117.3 | -15.4 | 5.9 |
| USGS-46 | -235.8 | -29.8 | 2.6 |
| South Pole Glacier Water | -434.5 | -54.3 | -0.1 |
| Custom Light Blend[†] | -573.7 | -76.2 | 36.1 |

*Standard values are reported relative to the VSMOW-SLAP scale

[†]The Custom Light Blend is a mixture of Purdue tap water, Sigma Aldrich deuterium depleted water (≤1 ppm HDO), and Isotec 95% $H_2^{18}O$ ($^{18}O$-enriched) to achieve a depleted isotopic signature that brackets the most depleted research flight observations of
δD and $δ^{18}O$ that also has a realistic d-excess signature. Because the Custom Light Blend is isotopically more depleted than our standards, known amounts of the Custom Light Blend and Purdue tap water were combined to make three mixtures, which were analysed using an LGR liquid water isotope analyser (T-LWIA-45-EP; model: 912-0050-0001) to determine the Custom Light Blend's isotopic signature.

The TWVIA's $H_2O_v$ dependence curve was reproducible over all $δ^{18}O$ isotope standard signatures considered (Table B1). The δD-$H_2O_v$ dependence curve was reproducible for the three relatively enriched isotope standards, more enriched than -235.8‰, in Table B1 but was not always reproducible using the most depleted standards (South Pole Glacier and Custom Light Blend) over the $H_2O_v$ range of ~3000 ppmv to ~ 8000 ppmv as is shown in the Figure B1a and B2a. At $H_2O_v$ mole fractions

above and below that range, the calibration curve remained reproducible. The cause of the 3000 ppmv – 8000 ppmv irreproducibility of the δD-$H_2O_v$ dependence curve associated with very depleted δD values remains unknown, perhaps small leaks in the experimental setup or uncertainty associated with curve fitting. To our knowledge this behaviour has not been described in the literature. However, δD values consistent with the two most depleted standards (Table B1) were only observed in the free troposphere and correspond to low $H_2O_v$ mole fractions (<1000 ppmv) and were outside of the irreproducible window

of $H_2O_v$ values. Therefore, it was not consequential to actual flight observations in this experiment. We note that there also appears to be large variability in the TWVIA-reported δD values <1000 ppmv $H_2O_v$ for the two depleted standards, but there is



also relatively larger variability in this $H_2O_v$ range for the enriched standards as well. To avoid biases resulting from the depleted δD irreproducibility, the δD water vapor dependence curve is defined using calibration data from the three relatively enriched standards. However, δD calibration data from each of the five standards is used to define uncertainties (see below in Appendix C).

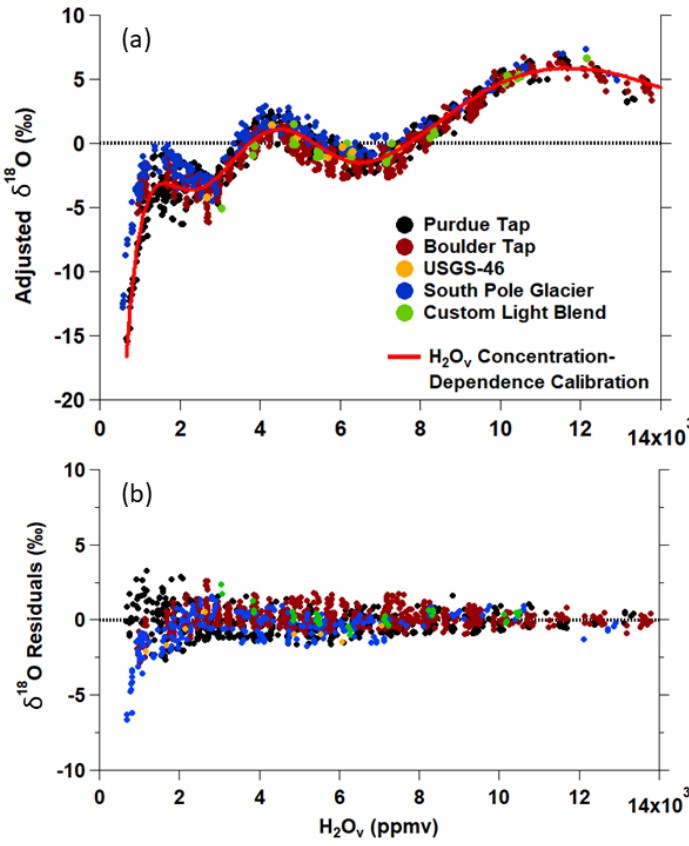

**Figure B1: δ<sup>18</sup>O-H₂Oᵥ dependence (a) calibration curve and (b) residuals. The true δ¹⁸O signature of each standard**
10 **(Table B1) has been subtracted from the TWVIA measurements to give the "adjusted" δ¹⁸O signature in (a).**



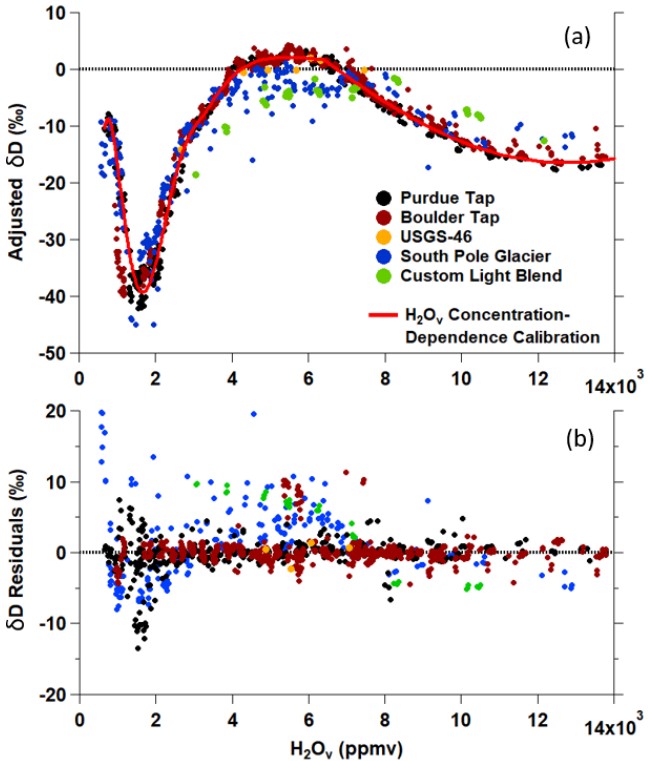

**Figure B2: δD-H₂Oᵥ dependence (a) calibration curve and (b) residuals. The true δD signature of each standard (Table B1) has been subtracted from the measurements to give the "adjusted" δD signature in (a).**

5        To check calibration the VSMOW-SLAP scale, Figure B3 shows that the linear regressions of the isotope standards' $H_2O_v$ concentration-dependence-corrected δ values versus true gas phase isotopic signature for (a) $\delta^{18}O$ and (b) δD have slopes near unity and intercepts near zero. $\delta^{18}O$ had a slope of 1.009($\pm$0.001), a y-intercept of 0.08($\pm$0.03), and an $R^2$ of 0.997254. The δD ordinary least squares regression line had a slope of 0.9954($\pm$0.0005), a y-intercept of -0.5($\pm$0.09), and an $R^2$ of 0.99958. A VSMOW-SLAP correction was not applied because it would be negligible compared to the uncertainty associated with the

10   concentration-dependence correction and the instrument precision (Appendix C).



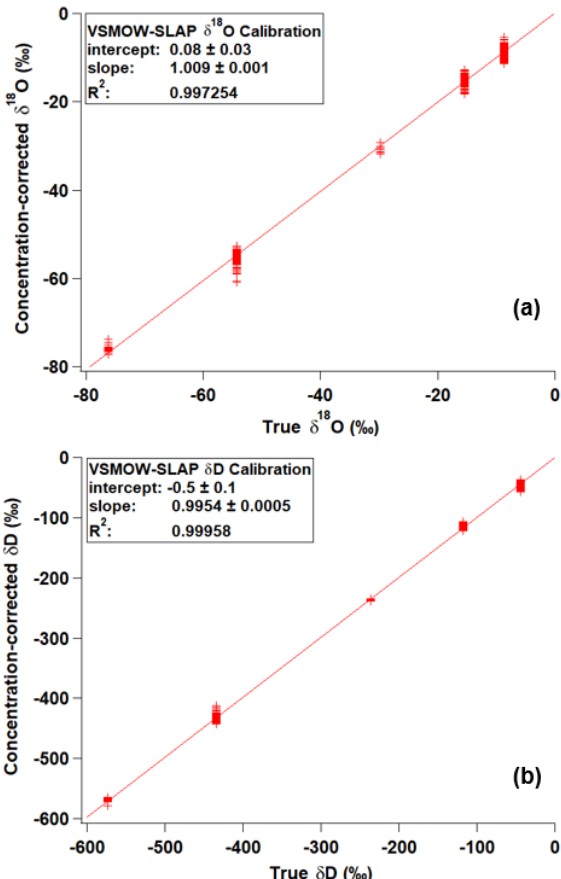

**Figure B3. VSMOW-SLAP calibration curves for (a) δ$^{18}$O and (b) δD. H$_2$O$_v$-concentration-corrected isotopic signatures are plotted against the standard's true isotopic signature. Linear regression fit slopes and intercepts are included in the figure insets.**

## Appendix C. Water vapor δD, δ$^{18}$O, and d-excess error propagation

**Instrument precision:**

10      The TWVIA instrument precision was calculated as the 1σ standard deviation for the last 20 seconds of every calibration period (Appendix B). The interval used to smooth the δD, δ$^{18}$O, and d-excess values reported in this paper is 20 s, which corresponds to the time required for the TWVIA signal to stabilize after a change in the sample's H$_2$O$_v$ mole fraction or isotopic signature. The δD and δ$^{18}$O precision values are calculated as a function of H$_2$O$_v$ mole fraction using power functions (Fig. C1).



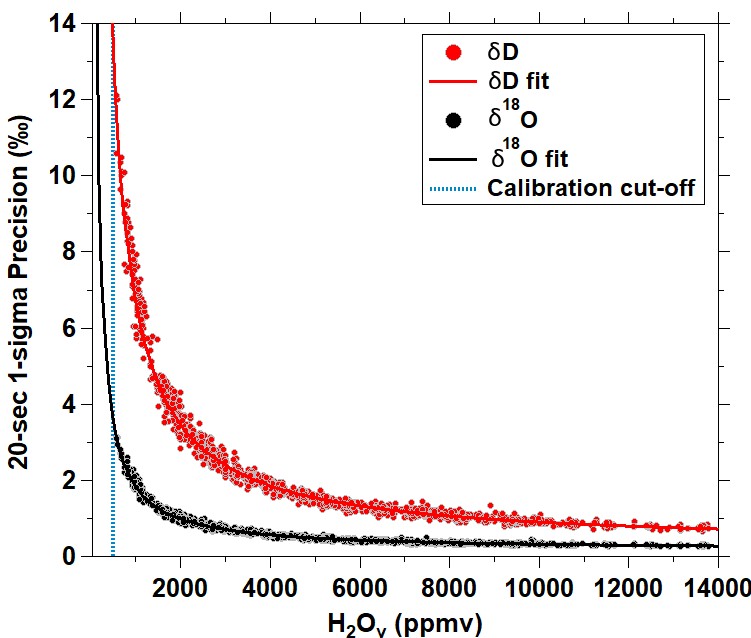

**Figure C1: TWVIA δ$^{18}$O and δD 20-s instrument precision (1σ) as a function of water vapor (H$_2$O$_v$) mole fraction.**

**H$_2$O$_v$ dependence calibration uncertainty:**

The uncertainty associated with the TWVIA δD- and δ$^{18}$O-H$_2$O$_v$ dependence corrections is determined from the calibration residuals shown in Fig. B1b and Fig. B2b. We note that the calibration residuals do include a small instrument precision component, as the calibration values are the average of 200 s sampling periods. The absolute value of the δD and δ$^{18}$O residuals from all five reference waters tested were filtered into bins defined by 100 ppmv H$_2$O$_v$ mole fraction increments. Averages of the absolute δD and δ$^{18}$O residuals were calculated for each bin. For relatively dry conditions (i.e. below 3500 ppmv

H$_2$O$_v$), the bin-averaged calibration residuals increase as H$_2$O$_v$ mole fractions decrease. A best-fit linear regression was determined for the bin-averaged residuals as a function of H$_2$O$_v$ mole fraction (from 550 – 3500 ppmv for δD and 550 – 3700 ppmv for δ$^{18}$O). Bin-averaged residuals were relatively constant for H$_2$O$_v$ mole fractions greater than 3500 ppmv for δD and 3700 ppmv for δ$^{18}$O. Average H$_2$O$_v$ dependence calibration uncertainties of 1.8‰ for δD and 0.9‰ for δ$^{18}$O were calculated from the bin-averaged residuals from 3500 – 14000 ppmv for δD and 3700 – 14000 ppmv for δ$^{18}$O. Higher uncertainties in the δ

values at low H$_2$O$_v$ mole fractions is not surprising, as the manufacturer suggests the TWVIA be used for sampling air ranging from 4,000 – 60,000 ppmv H$_2$O$_v$.

**Total uncertainty:**

      Total δD and δ$^{18}$O uncertainty is calculated by propagating the error resulting from instrument precision, $S_{precision}$, and

from the calibration, $S_{calibration}$, as in eq. (C1):

$$S_{total} = \sqrt{S_{precision}^2 + S_{calibration}^2}.$$      (C1)

The total d-excess uncertainty is determined according to eq. (C2):




$$S_{total,d-excess} = \sqrt{S_{total,\delta D}^2 + 8 \times (S_{total,\delta^{18}O}^2)}, \tag{C2}$$

where $S_{total,\delta D}$ and $S_{total,\delta^{18}O}$ are the total $\delta D$ and $\delta^{18}O$ uncertainties (given be eq. (C1)). The total uncertainty for $\delta D$, $\delta^{18}O$, and d-excess as function of $H_2O_v$ mole fraction is presented in Fig. 1.

**Appendix D. Fractionation of water vapor in ice supersaturated conditions**

$H_2O_v$ undergoing deposition in ice-supersaturated conditions is impacted by equilibrium and kinetic fractionation. The kinetic fractionation factor is calculated via Galewsky (2015) eq. (D1):

$$\alpha_{ice,k} = \frac{S_i}{\alpha_e \frac{D}{D'}(S_i-1)+1}, \tag{D1}$$

where $S_i$ is saturation with respect to ice, expressed as a fraction. The equilibrium fractionation factor $\alpha_e$ calculated for the

temperature at the lifting condensation level (LCL) and is discussed in Methods 2.5. The ratio of the molecular diffusivity of the light to heavy isotopologue, $\frac{D}{D'}$, is 1.02849 for $^{18}O$ and 1.02512 for D (Merlivat, 1978).

The isotopic signature of an air parcel in ice supersaturated conditions ($R_{S_i}$) can be calculated according to eq. (D2):

$$R_{S_i} = R_o \left( \frac{H_2O_v}{H_2O_{v_o}} \right)^{\alpha_{ice,k}\alpha_e - 1} \tag{D2}$$

$R_o$ is the heavy to light isotopologue ratio ($\frac{HDO}{H_2O}$ or $\frac{H_2^{18}O_v}{H_2O}$) of the parcel prior to the ascent. The remaining fraction of $H_2O_v$ left in

the ascending parcel relative to initial conditions is given by $\frac{H_2O_v}{H_2O_{v_o}}$.

Figure D1 shows the STC VP d-excess observations along with Raleigh vapor calculated from RH = 100 % (Methods 2.5) and vapor in ice supersaturated conditions (eq. (D2)). To match the most negative d-excess value observed at the top of the INV on STC, a supersaturation ($S_i$) of 1.17 (RH$_i$ = 117% in Fig. D1) was used but does not necessarily reflect reality for the temperature or altitude of the observations. The curve is presented for reference.





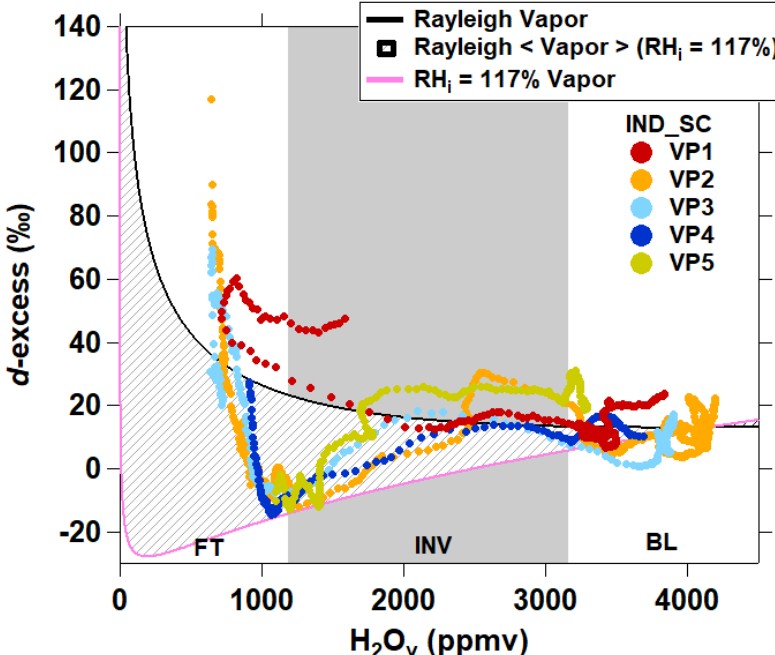

**Figure D1: STC VP d-excess observations, Rayleigh vapor d-excess, and calculated d-excess of vapor in ice supersaturated (RHi) conditions up to RHi = 117 %. Ice supersaturated conditions were chosen merely to match the INV-FT interface d-excess observations, and do not necessarily reflect a realistic $RH_i$ for the STC flight day.**

**Supplement link**

Figures S1-6 are provided in accompanying Supporting Information. Figure S3 and Figure S4 are GIFs available in separate files.

**Author contribution**

OS, LW, and PS designed the experiments. OS collected the airborne data, with the help of KH. OS analysed the data. OS, LW, MB, and PS interpreted the results. OS prepared the manuscript with contributions from all co-authors. BS maintained the experimental aircraft.

**Competing interests**

The authors declare no competing interests.

**Acknowledgments**

We thank Bruce Vaughn of the Institute of Arctic and Alpine Research (INSTAAR) for generously sharing a sample of South Pole Glacier isotope standard. We are grateful to Tomas Ratkus of Purdue University Science IT for his technological expertise while we prepared for the research flights. We thank LGR tech support for corresponding with us while we characterized the

20 TWVIA's water vapor concentration dependence. We thank Purdue University's Jonathon Amy Facility for Chemical Instrumentation (JAFCI) for their expertise in designing and maintaining ALAR's instrument suite. We acknowledge funding support from James Whetstone and the National Institute of Standards and Technology.





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
