# Peer review of "Vertical profile observations of water vapor deuterium excess in the lower troposphere"

_Atmospheric Chemistry and Physics, 2018_

## Referee Comment (RC1) · Anonymous Referee #1 · 3 Mar 2019

The authors present the analysis of airborne measurements of the isotopic composition of water vapour over the continental US. The study will be an interesting addition to the literature, and contributes a valuable dataset. Aspects of instrument uncertainty and data calibration are thoroughly presented. However, some aspects of the interpretation need further clarification, and the structure of the manuscript could be improved. I suggest major revisions of this manuscript, as detailed in the comments below.

[Figure]

**1  Major comments**

1. I am not convinced by the way the Rayleigh model is used as an explanation in flights RAY and STC. Even though a Rayleigh model can match much your data, it does not necessarily indicate that the guiding processes are correctly represented. In particular, the presence of a quite homogeneous mixing line of water vapour below the inversion, topped by a pronounced inversion layer, and relative humidity below 10% is in stark contrast to the moist adabatic profile that would be associated with a Rayleigh model. I recommend to include the possibility of long-range advection of air with subsequent mixing into the picture, as well as considering the role of subsidence of air as a contribution to the inversion layer. Can several mixing processes in time and in the vertical combined look like a Rayleigh curve? This analysis should be placed in the context of the critique of Taylor (1984) on Rozanski and Sonntag (1982), and the very valuable study by Gedzelman (1988) to analyze Taylor's (1972) data set. Maybe a conclusion of your study could be a critique rather than a statement of consistency with the Rayleigh model?

2. The interpretation of the stratocumulus case study as evaporating cloud drops needs further corroboration. The literature cited on the kinetic fractionation of evaporating droplets, such as Stewart (1975) considers rain drops, which are orders of magnitude larger, and have a vertical downward motion. This provides the potential of partial evaporation, leading to a kinetic fractionation signal in the d-excess. Cloud drops, however, will rapidly evaporate completely as they become unstable at smaller radii, leaving no sign of kinetic fractionation in the d-excess. Furthermore, they are suspended in the atmosphere, thus allowing for equilibrium fractionation. Without the consideration of rain evaporation in addition, or other processes, the explanation of the STC case is thus not viable. What is the potential of the ocurrence of rain evaporation from these stratocumulus clouds? Can

you estimate how much cloud water specific humidity would have to be added to produce the negative d-excess signal, and is this consistent with the saturation specific humidity in the cloud layer? Could there potentially have been a potential for rain evaporation or ice processes in the airmass upstream earlier instead?

3. Section 3.3 (DBL case) describes and interesting case of BL development, but is particularly difficult to follow. Consider reorganising the material in a more logical order, and formulate a clearer take-away in the end.

4. The material included in the Supplement only seemed marginally useful. It is also confusing to have both an appendix and a supplement in a manuscript. Consider merging the supplement into the Appendix, keeping only Fig. S1.

5. It is difficult to follow the description of the cases without additional context of the flight situations. Maybe Fig. 3 could be split up for each case, and supplemented by a weather chart similar to Fig. S6.

6. The discussion sections 4.3 and 4.4 appear repetetive to what has already been presented earlier an can be deleted.

7. The second half of the first paragraph of the Conclusions section should be moved to the introduction/motivation.

8. Important studies that should be referred to are Taylor (1972, 1984), and He and Smith (1999) and Ehhalt (1975, 2005) and Tsujimura et al., 2007.

9. I found the structure of the manuscript somewhat confusing, in that first the flights around both Washington DC and Indianapolis are presented, and then almost all analysis focuses on flights from only one site, before returning to a wider overview in section 3.4. It would be more logical for the reader to move Sec. 3.4 ahead of the case studies, and then zoom into individual aspects. Consider adding a more

intuitive way to present an overview of all data than a comparison to Rayleigh in 7.

10. Several times in the writing, wording such as "Fig. x shows ..." is used. For conciseness, consider rephrasing this to sentences talking about what is shown in the figure, added by a figure reference "(Fig. x)".

**2 Detailed comments**

P3, L8: "However, relatively few...": As far as I am aware there is only one published dataset of airborne d-excess measurements. Why the citation of Schmidt et al., 2005 here?

P3, L34: Please state if the inlet or parts of it were heated.

P4, L9: How sensitive are the measurements to variations of cavity pressure and temperature during flight?

P9, L21: Worden et al. 2007 formulate their models for rain evaporation (falling condensate), not cloud evaporation, see their supplementary information.

P10, L10: "Rayleigh-consistent observations": Consider rephrasing in light of major comment 1

Fig.4: Consider adding a panel that shows more mixing lines, e.g. between the bottom and top of the inversion, or the bottom and top of the BL, how do these compare with the complete mixing line?

P12, L1: Consider using a standard symbol such as $q$ or $m$ for specific humidity or mixing ratio of water vapour, rather than $H_2O$.

P14, L8: "relative to Rayleigh" rephrase as e.g. "Rayleigh model predictions"

P14, L16: "d-excess switches to tracking" rephrase

P14, L13-22: hard to follow, rewrite this section for clarity

P16, L6: "Differences" - difference to what is shown? Are these not absolute values?

Fig.6: How are RL and INV defined? You could provide a quantitative comparison of averaged quantities for both layers that supports their distinction.

P17, L14: Not clear what the take-away from this section is. Revise paragraph.

P17, L17: "Figure xxx shows": revise according to major comment 11

P18, L12: Not clear what the take-away from this section is. Revise paragraph.

P18, L16: Why the citation of Lee et al. (2006)? Compare Gedzelman (1988) and the discussion of the Taylor (1972) data (major comment 1).

P19, L3: The thermodynamic characteristics of the profiles are not consistent with a Rayleigh processes, even though the isotope composition may be - what is the conclusion from that finding?

P19, L15: How would cloud droplets be lofted but not the surrounding vapour? If vapour and droplets move together, how can non-equilibrium fractionation result?

P20, L16-20: Consider the possibility of rain evaporation upstream (major comment 2).

P21, L10: What relative roles could vertical motion and entrainment of dry air from aloft play in the evaporation of the cloud layer?

P21, L20-29: Rephrase for clarity. Could ice-phase processes have been relevant further upstream, earlier in time?

P22, L12: What is the Rapid Refresh Model? Could be deleted here.

P22, L7-34: reorganize and rewrite this section for clarity

P22, L36: "so we look" rephrase

Fig. A1: What is the reason for the large scatter between both instruments in the mid-range of humidities? Does longer averaging of the data provide a better agreement? Was the scatter similar for each of the flight? How is the scatter for the downward profiles only?

**3 References**

Ehhalt, D. H.: Vertical profiles of HTO, HDO, and H2O in the tro- posphere, National Center for Atmospheric Research, Boulder, Colorado, 1974.

Ehhalt, D. H., Rohrer, F., and Fried, A.: Vertical profiles of HDO/H2O in the troposphere, J. Geophys. Res., 110, D13301, doi:10.1029/2004JD005569, 2005.

Rozanski, K. and Sonntag, C.: Vertical distribution of deuterium in atmospheric water vapour, Tellus, 34, 135–141, 1982.

Rozanski, K. and Sonntag, C.: Reply to C. B. Taylor, Tellus B, 36, 71–72, doi:10.1111/j.1600-0889.1984.tb00054.x, 1984.

Taylor, C. B.: The vertical variations of isotopic concentrations of tropospheric water vapour over continental Europe, and their relationship to tropospheric structure, PhD thesis, Institute of Nuclear Sciences, Lower Hutt, New Zealand, 1972.

Taylor, C. B.: Vertical-distribution of deuterium in atmospheric water vapor: problems in application to assess atmospheric condensation models, Tellus B, 36, 67–70, 1984.

Tsujimura, M., Sasaki, L., Yamanaka, T., Sugimoto, A., Li, S.-G., Matsushima, D., Kotani, A., and Saandar, M.: Vertical distribu- tion of stable isotopic composition in atmospheric water vapor and subsurface water in grassland and forest sites, eastern Mon- golia, J. Hydrol., 333, 35–46, 2007.

---

## Referee Comment (RC2) · Anonymous Referee #2 · 12 Mar 2019

This paper describes vertical profiles of water vapor isotope ratios (e.g. dD, d18O) collected by aircraft in the lower atmosphere. The majority of the paper focuses on three case study flights, which took place around Indianapolis, USA. Additional flights were conducted both around Indianapolis and the Washington DC-Baltimore urban area, though much less is said about these. The paper pays particular attention to the utility of deuterium excess (dxs, defined as dD − 8×d18O) as a tracer of distinct air masses and of evaporating cloud condensate. The three case studies presented are purportedly illustrative of Rayleigh distillation, evaporation of stratocumulus clouds, and erosion of a residual layer as the boundary layer develops. The key points of the paper appear to be that Rayleigh distillation plays a dominant role in setting boundary layer water vapor isotope ratios and that deuterium excess is a useful tracer of mixing

between distinct air masses (and moisture sources) near the inversion layer.

The data collected are an important addition to the growing record of high-resolution water vapor isotope ratio measurements, and the attention to detail in the calibration and characterization of the measurements (reported in the Appendices) is quite impressive. The analysis also clearly demonstrates the utility of using deuterium excess to distinguish air masses, as compared to the individual isotope ratios. Nevertheless, I believe one of the main conclusions should be revisited and the clarity of the presentation improved.

Major Comments

1. The results could be better distilled. The paper is quite lengthy and detailed. As a result, it is easy for key points to become buried. Usually, I would not list this as a major concern, but in this case, I found much of the critical interpretation for the analysis was lost among the long-winded descriptions of the data. It might help readers if the critical arguments and conclusions were emphasized near the beginning and/or end of each sub-section. One place where this is very much needed is in Section 3.3, which discusses the developing boundary layer case (DBL). Here, more time could be spent on the interpretation of the causes of the fascinating differences in atmospheric structure among the profiles rather than on re-describing Figure 6.

2. Section 4.2 is another section that is somewhat difficult to follow, largely because of confusing terminology. How do the "scenarios" relate back to the equations presented in the methods? Also, "partial evaporation" and "near complete evaporation of a semi-dehydrated drop" could easily be confused as one and the same. As a result, it is not at all clear which "scenario" best represents the data.

3. I am not convinced by the analysis that Rayleigh distillation is the dominant process determining the vertical isotopic structure of the boundary layer. First and foremost, there are many papers that show the contrary; a few case studies are not sufficient to prove otherwise. Previous papers that have measured water vapor isotope ratio

vertical profiles in situ (and have shown profiles consistent with processes other than distillation) include He and Smith 1999, Galewsky et al 2007, Noone et al 2013, Bailey et al 2013, Sodemann et al 2017, and Kelsey et al 2018. These studies contrast, to some degree, with the early work of Ehhalt, which was re-published as Ehhalt et al 2005.

Second, I am not entirely convinced that Rayleigh distillation gives the best physical interpretation for the Indianapolis "Ray" profiles. The paper argues that Rayleigh distillation is a good model when the boundary layer is dry adiabatic (and therefore that no clouds or precipitation exist). This is contrary to expectation: distillation depends on condensation and precipitation. Furthermore, other studies (see above) have shown nearly the opposite: that Rayleigh is appropriate when the boundary layer follows a moist adiabat but not when it follows a dry one. A clear exception, of course, is if the distillation occurs upwind and imprints an isotopic signature that is then advected downwind. One of the earliest papers discussing this phenomenon is Gedzelman 1988. Is it possible that advection is affecting the Indianapolis isotope ratio profiles? If so, this could make for an interesting discussion on whether moist convective processes regionally set the humidity structure of the lower atmosphere locally, which others have argued for tropical/subtropical regions (e.g. Brown et al 2008, Lee et al 2011, Bailey et al 2013).

Third, extra care must be taken in matching data to hypothetical Rayleigh curves since these can be designed to fit many data. A good example of this is found in Noone et al 2013. Consequently, it may be difficult to truly distinguish Rayleigh from mixing processes unless the theoretical end-members are well constrained. It is not clear in the manuscript from whence the theoretical end-members for Figure 4 are derived. Some description of the assumptions made would be a valuable addition to the analysis and perhaps make the case that the Ray flights are, in fact, illustrative of (upwind?) Rayleigh distillation in a much more compelling manner.

4. Some care should be taken in describing the Worden et al 2007 and Stewart 1975

models and applying them to the case of stratiform clouds. Both models were designed to describe freely falling raindrops. In the original presentation of the model, Worden suggests raindrops undergo both an equilibrium fractionation and an effective fractionation, and that "this assumption is unlikely to be valid when raindrops are small...". I note that Equation 3 substitutes a kinetic fractionation coefficient in place of the effective factor. What impact does this have? How does the model work if the assumptions of large falling drops are violated? The Worden model describes isotopic depletion with a loss of water from an air parcel. How can it be applied to describe a gain of moisture by the atmosphere? I have similar concerns with use of the Stewart model and would like to see more justification for these model choices for the case of stratiform clouds. Also, the equations presented from Stewart are from Equations 2 and 3 of the original paper, and there is an alpha missing in the denominator of the beta equation.

5. I found it difficult to identify the case studies in the flight figures due to distinct nomenclature. The figures use numbers, the text uses pseudonyms, and only the table provides both these plus dates. I would recommend one naming convention, preferably related to flight number or date. The reason being that a priori, it seems difficult to know whether the "Ray" days will really be Rayleigh-like.

6. The Isotope Theory section suggests there are "three common ways the isotopic composition of the atmospheric $H_2Ov$ can change." I would have thought these would be condensation, evaporation, and mixing. Rayleigh distillation is just an example model for condensation processes. Similarly, cloud evaporation is just one type of evaporation that can affect the atmosphere's isotopic composition.

7. I have some trouble understanding how partial cloud evaporation can cause a minimum of deuterium excess near the inversion layer. Evaporation tends to favor the diffusion of the D relative to 18O. Why wouldn't partial evaporation result in an enrichment of the surrounding vapor? Perhaps I am missing something, but my hunch is most readers will also have this impression. It might be worth explaining the physical underpinning behind these conclusions in greater depth, perhaps in Sections 4.2 or

4.4.

8. The calibration documentation is quite thorough and comprehensive. I was prompted, however, to ask a few follow up questions regarding the variations in concentration dependence shown for dD . Is it possible one would get a different answer if concentration biases were adjusted first and VSMOW-scaling applied separately? It also appears that there are higher errors in dD at low isotopic values at all water vapor concentrations, not just at the low concentrations. Is it possible that lower precision at low isotope ratios causes the appearance of "irreproducibility" in the concentration dependence?

Minor comments

Page 1 Line 35 – no need for ":" after "include"

Page 2 Line 10 –I had trouble distinguishing the conditions at a moisture source region from "surface H2Ov sources." Perhaps it might be more clear to say "an air parcel's moisture source region, including the geography of the source and its meteorological conditions?"

Page 2 Line 19 – I think "further exchange" is meant instead of "equilibrium?"

Page 3 Line 3 – I might remove "point in" before time, since I initially confused "point" with space.

Page 3 Line 6 – Perhaps best to say "higher…resolution" since aircraft is not as high resolution as slower-moving platforms.

Page 3 Line 11 – Perhaps best to say that "measurements of vertical profiles" were conducted.

Page 4 Line 28 – Perhaps "produce" for "emit?"

Page 6 Figure 2 – Could the three analyzed flights be emphasized, perhaps by making the other flight lines dashed?

Page 7 Table 1 – Table caption/title should provide some explanation of the "codes" and what is meant by "support study"

Page 9 Line 13 – One of many examples of the great care that was taken in the analysis

Page 9 Line 31 – This appears to be the only place where "q" is used instead of "H2Ov." Consistency would help.

Page 10 Equations – This appears to be the only place "Rvap" is used instead of "Rv." Again, consistency would help.

Page 11 Figure 4 – All the lines are "solid," thus it might be best to say the "black" line to distinguish it from the "pink" one. Also, I don't fully understand how the average mixing ratio is given by a gray envelope. Shouldn't the average be a point?

Page 12 Line 7+ - Here is an example where it's easy for the reader to become lost in all the number-reporting. This paragraph would greatly benefit from a sentence that provides a bit more of a picture of what is going on physically.

Page 12 Line 18 – I might say "predictions" instead of "theory."

Page 13 Figure 5 – Caption should explain what the shaded area for the inversion is and what the envelopes around the profiles are.

Page 14 Line 6 – How was the average range of the inversion calculated? From how many days or which days of data?

Page 14 Line 16 – I'm not sure I agree that the data start "tracking" the mixing line. There just aren't enough points for me to be convinced of that. Perhaps "approaches" the mixing line?

Page 14 Line 26 – This seems like an important argument explaining how is STC different from Ray, and yet it is buried halfway down the page. Perhaps it could be moved up in the sub-section.

Page 15 Figure 6 – Most axes appear consistent across panels except for theta. Was this purposeful?

Page 17 subtitle – I'm not really sure what "general observations" means. Would "observations from other flights" be more descriptive?

Page 17 Line 21 – which observations are used here for this argument?

Page 19 Line 19 – A sentence or couple words reminding the reader what "Scenario 1" is would be appreciated.

Page 19 Line 22 – I think "drier" is meant.

Page 21 Figure 9 – I would recommend dots (or some symbol) instead of vertical lines to indicate the values of dxs expected. Otherwise, it is not clear that the reader should look for the intersection of the various lines.

Page 22 Line 24 – Excellent synthesis sentence: highlights the key point nicely.

Page 22 Line 36 – I disagree that these are some of the few data of this kind. There are quite a few studies that are not cited in this work. Please see my major comments for ideas.

Page 23 Line 24 – Kelsey et al 2018 also report dxs profiles.

Page 24 Line 4 – Perhaps "investigate" for "interrogate."

Page 27 Line 5 – "To check calibration. . ." against? of?

Page 30 Line 14 – The notation seems to change here. Should all isotopologues have subscript "v?"

Page 31 Figure D1 – At first I thought the black square was a symbol in the legend. It might be more clear simply to use the caption to say that the striped region shows the expected range for the observations, or something to that effect.

---

## Author Comment (AC1) · 9 May 2019

See attached pdf for formatted version with author responses in color.

Authors' Responses to Reviewer 1 • Author responses are indicated in blue font. • Locations of our edits in the "strike-through" and final versions of the manuscript are provided with the following convention: ([strike-through version] pg. #, ln # / [final version] pg. #, ln #).

1 Major comments

1. I am not convinced by the way the Rayleigh model is used as an explanation in flights RAY and STC. Even though a Rayleigh model can match much your data, it does not necessarily indicate that the guiding processes are correctly represented. In

particular, the presence of a quite homogeneous mixing line of water vapour below the inversion, topped by a pronounced inversion layer, and relative humidity below 10% is in stark contrast to the moist adiabatic profile that would be associated with a Rayleigh model. I recommend to include the possibility of long range advection of air with subsequent mixing into the picture, as well as considering the role of subsidence of air as a contribution to the inversion layer. Can several mixing processes in time and in the vertical combined look like a Rayleigh curve? This analysis should be placed in the context of the critique of Taylor (1984) on Rozanski and Sonntag (1982), and the very valuable study by Gedzelman (1988) to analyze Taylor's (1972) data set. Maybe a conclusion of your study could be a critique rather than a statement of consistency with the Rayleigh model? The Reviewer makes an excellent point that our flight conditions during the RAY (now CLR) flights are not consistent with moist adiabatic conditions that would be required for Rayleigh rainout processes to be actively occurring. However, the Rayleigh prediction is most consistent with our observations, compared to mixing scenarios as we now show in Figure 7. We have changed this discussion to indicate that observed profile is a fingerprint of previous airmass dehydration conditions that is retained by transport and downward mixing of dehydrated higher altitude FT air, consistent with past studies (Taylor, 1984; Gedzelman, 1988). (pg 19, ln 2-28 / pg 8, ln 10-33)

2. The interpretation of the stratocumulus case study as evaporating cloud drops needs further corroboration. The literature cited on the kinetic fractionation of evaporating droplets, such as Stewart (1975) considers rain drops, which are orders of magnitude larger, and have a vertical downward motion. This provides the potential of partial evaporation, leading to a kinetic fractionation signal in the dexcess. Cloud drops, however, will rapidly evaporate completely as they become unstable at smaller radii, leaving no sign of kinetic fractionation in the d-excess. Furthermore, they are suspended in the atmosphere, thus allowing for equilibrium fractionation. Without the consideration of rain evaporation in addition, or other processes, the explanation of the STC case is thus not viable. What is the potential of the occurrence of rain evaporation from these stratocumulus clouds? Can you estimate how much cloud water specific humidity would have to be added to produce the negative d-excess signal, and is this consistent with the saturation specific humidity in the cloud layer? Could there potentially have been a potential for rain evaporation or ice processes in the airmass upstream earlier instead? The Reviewer again brought up some excellent details to consider. Our calculations using the equations in Stewart likely apply to cloud droplets evaporating just like rain droplets. Similarly, we realize that the assumptions of the Worden equations are not valid either because that is for a case of a dehydrating airmass (closed system Rayleigh). However, there are 2 important timescales to consider when exploring the influence of liquid droplet evaporation in the inversion layer. (1) The timescale at which a liquid droplet isotopically equilibrates with its surrounding vapor. (2) The speed at which droplets move through the inversion layer. We calculated the first timescale using Eqn. B5 is Bolot et al., 2013 which is ~2 seconds for a cloud droplet size of 15 microns. For our observations of vertical wind speeds, the time for a droplet to move from the bottom to the top of the inversion is 19 seconds. These calculations do indicate that cloud droplet evaporation starting in the lower inversion and finishing in the top of the inversion during the flight conditions in unlikely. However, if previous inversion conditions were colder, if the droplets were larger than 50 microns (like drizzle), or if the inversion had faster vertical wind speeds, these values could converge. We have edited the discussion of the STC flight day to reflect these new considerations. (pg 24, ln 19 – pg 25, ln 39 / pg 10, ln 17 – pg 11, ln 2)

3. Section 3.3 (DBL case) describes and interesting case of BL development, but is particularly difficult to follow. Consider reorganising the material in a more logical order, and formulate a clearer take-away in the end. We have consolidated the DBL case study's results (originally Sect. 3.3) and discussion sections (originally Sect 4.3) so that the case study's key isotope features are identified, and their cause(s) immediately discussed. Furthermore, we have extensively revised Section 3.3 (now Section 4.3), which presents the DBL case study results.

4. The material included in the Supplement only seemed marginally useful. It is also confusing to have both an appendix and a supplement in a manuscript. Consider merging the supplement into the Appendix, keeping only Fig. S1. This is a good suggestion. We have merged the Appendix and the important components of the Supplemental Information (SI) into the SI in order to further limit the length of the paper.

5. It is difficult to follow the description of the cases without additional context of the flight situations. Maybe Fig. 3 could be split up for each case, and supplemented by a weather chart similar to Fig. S6. We have split up Fig 3a-c to make three plots for the three case studies (Fig 5, 8, 10) so that they can be placed closer to the text where they are being discussed. Weather charts have been added to the SI for each case studies (Fig. S5.1 – S5.3).

6. The discussion sections 4.3 and 4.4 appear repetitive to what has already been presented earlier and can be deleted. We have consolidated the Results and Discussion sections so that discussion paragraphs directly follow their respective results sections. This reorganization has reduced repetitive parts of discussion sections 4.3 and 4.4. We have also reorganized the manuscript so that discussion of the campaign-wide observations (previously Sect 4.4, now Sect 3) now precedes discussion of the case study observations.

7. The second half of the first paragraph of the Conclusions section should be moved to the introduction/motivation. We have made the suggested change.

8. Important studies that should be referred to are Taylor (1972, 1984), and He and Smith (1999) and Ehhalt (1975, 2005) and Tsujimura et al., 2007. We did not focus on these studies because our primary interest was high-frequency deuterium-excess observations, but the reviewer is right that they provide great context for our work. This is why we now point to the the detailed overview of airborne water vapor isotope studies reviewed in the introduction of Sodemann et al., 2017 (pg. 3, ln. 37/ pg. 3, ln. 9).

9. I found the structure of the manuscript somewhat confusing, in that first the flights

around both Washington DC and Indianapolis are presented, and then almost all analysis focuses on flights from only one site, before returning to a wider overview in section 3.4. It would be more logical for the reader to move Sec. 3.4 ahead of the case studies, and then zoom into individual aspects. Consider adding a more intuitive way to present an overview of all data than a comparison to Rayleigh in 7. This is a great suggestion. We have followed the Reviewer's suggestion by reorganizing the manuscript so that the campaign-wide results/discussion precede the case studies' results/discussion sections. We have added a new figure (Fig. 3) which shows dD and d18O vs H2Ov along every VP descent conducted during the campaign. We no longer present our results relative to Rayleigh by removing the right panel of Fig 7 (now Fig 4 post reorganization).

10. Several times in the writing, wording such as "Fig. x shows ..." is used. For conciseness, consider rephrasing this to sentences talking about what is shown in the figure, added by a figure reference "(Fig. x)". This is a good suggestion, we have made changes where appropriate.

2 Detailed comments P3, L8: "However, relatively few...": As far as I am aware there is only one published dataset of airborne d-excess measurements. Why the citation of Schmidt et al., 2005 here? We have reworded the sentence and deleted the reference to Schmidt et al., 2005, which provides modeled d-excess.

P3, L34: Please state if the inlet or parts of it were heated. We have added a sentence stating the TWVIA inlet was not heated during the calibrations. (pg 5, ln 27-28 / pg 4, ln 35-36)

P4, L9: How sensitive are the measurements to variations of cavity pressure and temperature during flight? Once the isotope analyzer has warmed up (i.e. reached the manufacturer recommended pressure and temperature values for operation), cavity pressure and temperature only vary ($1\sigma$) by $\pm0.02$ Torr and $\pm0.08$oC, respectively, over a vertical profile descent on average. This is within the operating specification

given by the manufacturer. (pg. 5 ln 13-15 / pg 4, ln 21-22)

P9, L21: Worden et al. 2007 formulate their models for rain evaporation (falling condensate), not cloud evaporation, see their supplementary information. We have removed all discussion of the Worden et al., 2007 model (see our response to major comment #2 for more explanation).

P10, L10: "Rayleigh-consistent observations": Consider rephrasing in light of major comment 1 This is a great suggestion. We have renamed the RAY case study to CLR, which is now representative of the case study's clear sky conditions rather than a possible interpretation of the day's dominant isotopic process.

Fig.4: Consider adding a panel that shows more mixing lines, e.g. between the bottom and top of the inversion, or the bottom and top of the BL, how do these compare with the complete mixing line? We have split up Fig. 4 into three figures for the three case studies (Fig. 7 CLR, Fig. 9 STC, and Fig 12 DBL). Each new case study figure has an additional panel showing mixing lines for scenarios relevant to the flight day. The new mixing panels are discussed in each case study's respective sections (Sect 4.1-4.3).

P12, L1: Consider using a standard symbol such as q or m for specific humidity or mixing ratio of water vapour, rather than H2O. We respectfully choose to retain H2Ov to represent water vapor mole fraction, as it is a common convention in the atmospheric chemistry field.

P14, L8: "relative to Rayleigh" rephrase as e.g. "Rayleigh model predictions" The sentence now reflects this suggestion.

P14, L16: "d-excess switches to tracking" rephrase This sentence has been rephrased.

P14, L13-22: hard to follow, rewrite this section for clarity We thank the Reviewer for identifying this paragraph that could be improved. We have revised this paragraph to include a summarizing, take-away sentence, which highlights the unique d-excess characteristics of the first vertical profile flown on STC (pg 23, ln 17-20 / pg 9, ln 36-40).

P16, L6: "Differences" - difference to what is shown? Are these not absolute values? This sentence has been rephrased.

Fig.6: How are RL and INV defined? You could provide a quantitative comparison of averaged quantities for both layers that supports their distinction. To address this comment, we have added a sentence that reads, "We define the base of the RL using the same approach described in Section 2.4 ($d\theta/dz$ and $|d(H_2Ov)|/dz$ threshold values) for determining the base of the INV ($z_{INV}$)."

P17, L14: Not clear what the take-away from this section is. Revise paragraph. We thank the Reviewer for identifying this paragraph as an opportunity for improvement. We have revised this paragraph about our campaign-wide vertical profile observations. (Sect. 3, paragraph 1-2)

P17, L17: "Figure xxx shows": revise according to major comment 11 We have re-worded sentences that begin with "Figure # shows…" throughout the manuscript.

P18, L12: Not clear what the take-away from this section is. Revise paragraph. We have deleted this paragraph because we decided its inclusion does not benefit the overall story about the campaign-wide observations section.

P18, L16: Why the citation of Lee et al. (2006)? Compare Gedzelman (1988) and the discussion of the Taylor (1972) data (major comment 1). During our consolidation of the results and discussion sections for each case study, we have removed the indicated sentence. We now discuss the possibility of Rayleigh-consistent condensation occurring upwind of the CLR measurements (see response to major comment #1). We also reference the suggested paper (pg 19, ln 17-28 / pg 8, ln 22-33)

P19, L3: The thermodynamic characteristics of the profiles are not consistent with a Rayleigh processes, even though the isotope composition may be – what is the conclusion from that finding? We now discuss the possibility of Rayleigh-consistent condensation occurring upwind of the CLR measurements (see response to major comment

**1). (pg 19, ln 17-28 / pg 8, ln 22-33)**

P19, L15: How would cloud droplets be lofted but not the surrounding vapour? If vapour and droplets move together, how can non-equilibrium fractionation result? The vapor and droplets do not necessarily have to remain in isotopic equilibrium under changing conditions, including mixing of airmasses. We have revised the indicated sentence so that cloud droplet evaporation is discussed from the framework of timescales, e.g. the time required for condensate and vapor to isotopically equilibrate vs. time required to transport a cloud droplet from the bottom to the top of the inversion. Please see our response to major comment #2 for more detail. (pg 24, ln 19 – pg 25, ln 39 / pg 10, ln 17 – pg 11, ln 2)

P20, L16-20: Consider the possibility of rain evaporation upstream (major comment 2). We now discuss how rain droplet evaporation in dry, cold conditions upwind of the STC flight measurements could possibly explain the minimum in d-excess at the top of the inversion layer. (pg 25, ln 33-36 / pg 10, ln 36-38)

P21, L10: What relative roles could vertical motion and entrainment of dry air from aloft play in the evaporation of the cloud layer? This is an interesting question that we believe that stable isotope observations can inform about the occurrence of cloud evaporation, but are not sure that it's possible to distinguish the driver of that evaporation (cloud drops moving into the FT versus entrainment of dry air into the cloud) using our observations downwind of clouds. This would require a sophisticated cloud microphysics model, similar to Bolot et al. (2013). Respectfully, it is outside of the scope of this paper, but an exciting future application. We do, however, cite papers that give evidence for stratocumulus cloud droplet evaporation occurring within and above the cloud layer due to differences in entrainment efficiency (pg 24, ln 36- pg 25 ln 2 / pg 10, ln 12-15).

P21, L20-29: Rephrase for clarity. Could ice-phase processes have been relevant further upstream, earlier in time? We have rephrased the paragraph about the effect condensation in the presence of ice has on vapor d-excess for clarity. We also now include the following sentence, "It is possible, however, that condensation under ice-supersaturated conditions occurred prior to the STC flight, and that the resulting isotopic imprint was maintained during transport to Indianapolis." (pg 26, ln 28-30 / pg 11, ln 15-17)

P22, L12: What is the Rapid Refresh Model? Could be deleted here. Reference to the model name has been deleted.

P22, L7-34: reorganize and rewrite this section for clarity We have revised the entire results/discussion sections related to the DBL case study to improve their clarity. (now Sect 4.3) P22, L36: "so we look" rephrase This phrase has been deleted as part of the manuscript reorganization.

Fig. A1: What is the reason for the large scatter between both instruments in the mid-range of humidities? Does longer averaging of the data provide a better agreement? Was the scatter similar for each of the flight? How is the scatter for the downward profiles only? The scatter corresponds to few points relative to most of the points that show good agreement. The mid-range humidities in Fig. A1 (now Fig. S1 due to the Appendix-SI merger) correspond to vertical profiles. These few points that do not show as good agreement result from the TWVIA data being low pass filtered relative to the Picarro due to a longer residence time, as we mention in Sect 2.2.2 (SI pg 5, ln 23 / SI pg 4, ln 31).

Please also note the supplement to this comment:
https://www.atmos-chem-phys-discuss.net/acp-2018-1313/acp-2018-1313-AC1-supplement.pdf
* * *

---

## Author Comment (AC2) · 9 May 2019

See attached pdf for formatted version with author responses in color.

Authors' Responses to Reviewer 2 • Author responses are indicated in blue font. • Locations of our edits in the "strike-through" and final versions of the manuscript are provided with the following convention: ([strike-through version] pg. #, ln # / [final version] pg. #, ln #).

Major Comments

1. The results could be better distilled. The paper is quite lengthy and detailed. As a result, it is easy for key points to become buried. Usually, I would not list this as a major concern, but in this case, I found much of the critical interpretation for the analysis

was lost among the long-winded descriptions of the data. It might help readers if the critical arguments and conclusions were emphasized near the beginning and/or end of each sub-section. One place where this is very much needed is in Section 3.3, which discusses the developing boundary layer case (DBL). Here, more time could be spent on the interpretation of the causes of the fascinating differences in atmospheric structure among the profiles rather than on re-describing Figure 6. To address the Reviewer's comment, we have consolidated the case studies' results (originally Sect. 3.1-3.3) and discussion sections (originally Sect 4.1-4.3) so that each case study's key isotope features are identified, and their possible cause(s) immediately discussed. The consolidated sections are now located in Section 4.1-4.3. We have also extensively revised Section 3.3 (now Section 4.3), which presents the DBL case study results.

2. Section 4.2 is another section that is somewhat difficult to follow, largely because of confusing terminology. How do the "scenarios" relate back to the equations presented in the methods? Also, "partial evaporation" and "near complete evaporation of a semidehydrated drop" could easily be confused as one and the same. As a result, it is not at all clear which "scenario" best represents the data. We thank the Reviewer for identifying Section 4.2 as an area that could see improvement. We have chosen to remove the cloud evaporation equations (see our response to comment #4), and by association, the paragraphs which discussed the degree of droplet evaporation under two scenarios. In their place, we have added text which evaluate the likelihood of cloud evaporation and its impact on d-excess under meteorologically-relevant time scales. We believe our changes have greatly improved the clarity of Section 4.2. (pg 24, ln 19 – pg 25, ln 39 / pg 10, ln 17 – pg 11, ln 2)

3. I am not convinced by the analysis that Rayleigh distillation is the dominant process determining the vertical isotopic structure of the boundary layer. First and foremost, there are many papers that show the contrary; a few case studies are not sufficient to prove otherwise. Previous papers that have measured water vapor isotope ratio vertical profiles in situ (and have shown profiles consistent with processes other than

distillation) include He and Smith 1999, Galewsky et al 2007, Noone et al 2013, Bailey et al 2013, Sodemann et al 2017, and Kelsey et al 2018. These studies contrast, to some degree, with the early work of Ehhalt, which was re-published as Ehhalt et al 2005. We thank the Reviewer for these suggested literature citations. We have revised our interpretation of the RAY (now CLR) case study to express that the Rayleigh-consistent observations likely reflect the isotopic imprint of prior condensation under saturated conditions, followed by advection of this imprinted signal to the Indianapolis study site. Additionally we note that the lower free troposphere measurements appear more consistent with mixing lines, which is indicative of mixing between subsiding free troposphere air and boundary layer air. We include the some of the suggested references to support this discussion (pg 19, ln 2-28 / pg 8, ln 10-33). Second, I am not entirely convinced that Rayleigh distillation gives the best physical interpretation for the Indianapolis "Ray" profiles. The paper argues that Rayleigh distillation is a good model when the boundary layer is dry adiabatic (and therefore that no clouds or precipitation exist). This is contrary to expectation: distillation depends on condensation and precipitation. Furthermore, other studies (see above) have shown nearly the opposite: that Rayleigh is appropriate when the boundary layer follows a moist adiabat but not when it follows a dry one. A clear exception, of course, is if the distillation occurs upwind and imprints an isotopic signature that is then advected downwind. One of the earliest papers discussing this phenomenon is Gedzelman 1988. Is it possible that advection is affecting the Indianapolis isotope ratio profiles? If so, this could make for an interesting discussion on whether moist convective processes regionally set the humidity structure of the lower atmosphere locally, which others have argued for tropical/subtropical regions (e.g. Brown et al 2008, Lee et al 2011, Bailey et al 2013). The reviewer make an excellent point that our fight conditions during the RAY (now CLR) flights are not consistent with moist adiabatic conditions. However, the Rayleigh prediction is most consistent with our observations, compared to mixing scenarios. We have changed this discussion to indicate that observed profile is an imprint of previous airmass dehydration conditions, and cite the suggested studies (pg 19, ln 17-28 / pg

8, ln 22-33). Third, extra care must be taken in matching data to hypothetical Rayleigh curves since these can be designed to fit many data. A good example of this is found in Noone et al 2013. Consequently, it may be difficult to truly distinguish Rayleigh from mixing processes unless the theoretical end-members are well constrained. It is not clear in the manuscript from whence the theoretical end-members for Figure 4 are derived. Some description of the assumptions made would be a valuable addition to the analysis and perhaps make the case that the Ray flights are, in fact, illustrative of (upwind?) Rayleigh distillation in a much more compelling manner. The Reviewer makes a good point. We do note, however, that our Rayleigh curves are calculated using an objective method (Sect. 2.5; Eq (1)). The initial isotopic composition of the Rayleigh air parcel (Ro) is determined from the average observed boundary layer delta values. The equilibrium fractionation factors are calculated for the lifting condensation level temperature. We also demonstrate in Fig. S4 that varying the equilibrium fractionation factor by observed temperature does not lead to significantly different shaped Rayleigh curves. We now note that mixing endmembers are informed by actual observed delta values in different atmospheric layers (pg 10, ln 26 / pg 6, ln 26). Figure 7 now makes a direct comparison of our observations to Rayleigh theory and mixing scenarios.

4. Some care should be taken in describing the Worden et al 2007 and Stewart 1975 models and applying them to the case of stratiform clouds. Both models were designed to describe freely falling raindrops. In the original presentation of the model, Worden suggests raindrops undergo both an equilibrium fractionation and an effective fractionation, and that "this assumption is unlikely to be valid when raindrops are small...". I note that Equation 3 substitutes a kinetic fractionation coefficient in place of the effective factor. What impact does this have? How does the model work if the assumptions of large falling drops are violated? The Worden model describes isotopic depletion with a loss of water from an air parcel. How can it be applied to describe a gain of moisture by the atmosphere? I have similar concerns with use of the Stewart model and would like to see more justification for these model choices for the case of stratiform clouds. Also, the equations presented from Stewart are from Equations 2 and 3 of the original paper, and there is an alpha missing in the denominator of the beta equation. We have opted to remove the Worden et al, 2007 and Stewart, 1975 equations from the paper for the reasons that both reviewers mention. Instead of calculating the impact different degrees of evaporation could have on surrounding vapor, we have reframed this discussion by evaluating the likelihood of cloud droplet evaporation impacts based on transport and equilibration time scales. We now show that a droplet would isotopically equilibrate with surrounding vapor faster than the time required for transport of a droplet from the bottom to the top of the inversion (Equation B5 in Bolot et al., 2013). These calculations suggest cloud droplet evaporation may not be responsible for the observed d-excess minimum during the STC flight. However, we maintain that a negative d-excess signal resulting from cloud or rain droplet evaporation could have been transported from an area upwind of the STC measurements. (pg 24, ln 19 – pg 25, ln 39 / pg 10, ln 17 – pg 11, ln 2)

5. I found it difficult to identify the case studies in the flight figures due to distinct nomenclature. The figures use numbers, the text uses pseudonyms, and only the table provides both these plus dates. I would recommend one naming convention, preferably related to flight number or date. The reason being that a priori, it seems difficult to know whether the "Ray" days will really be Rayleigh-like. The Reviewer brings up an excellent point. We have deleted any reference to research flight codes (RF#), and now refer to the flights by date. We maintain pseudonyms for the case study flights, but we have made sure that figures include both the flight date and case study pseudonym where appropriate. We have renamed the RAY case study to CLR (for clear skies) so that the pseudonym characterizes the day's meteorology, rather than a possible interpretation of the isotopologue data.

6. The Isotope Theory section suggests there are "three common ways the isotopic composition of the atmospheric H2Ov can change." I would have thought these would be condensation, evaporation, and mixing. Rayleigh distillation is just an example model for condensation processes. Similarly, cloud evaporation is just one type of

evaporation that can affect the atmosphere's isotopic composition. We have reworded this section to indicate that we employ different models to represent condensation and mixing processes. Please refer to our response to comment #4 for our explanation for removing the cloud evaporation equations. (pg 9, ln 24-26 / pg 6, ln 1-4)

7. I have some trouble understanding how partial cloud evaporation can cause a minimum of deuterium excess near the inversion layer. Evaporation tends to favor the diffusion of the D relative to 18O. Why wouldn't partial evaporation result in an enrichment of the surrounding vapor? Perhaps I am missing something, but my hunch is most readers will also have this impression. It might be worth explaining the physical underpinning behind these conclusions in greater depth, perhaps in Sections 4.2 or 4.4. The Reviewer is correct that evaporation (from an infinitely large source) typically imparts a positive d-excess signal on the surrounding vapor. This is why we believe cloud evaporation to be responsible for the slight increase in d-excess in the middle of the inversion (Fig. 6b). As a droplet evaporates, its own d-excess signal becomes more negative, so subsequent complete evaporation of the droplet in another region would act to decrease the surrounding vapor d-excess (Aemisegger et al., 2015; Sodemann et al., 2017). We also now include a calculation (Bolot et al., 2013) showing that a droplet would isotopically equilibrate with surrounding vapor faster than the time a droplet would be transported from the bottom of the inversion to the top of the inversion. This calculation indicates cloud evaporation during the STC flight may not be responsible for the observed d-excess minimum. We do, however, maintain that evaporation of cloud/rain droplets upwind might have caused the d-excess minimum. We support this possible explanation with references that show that stratocumulus cloud droplet evaporation occurs at different altitude in and above the cloud layer, and that the inversion layer above stratocumulus clouds are not homogeneous in terms of depth or thermodynamic properties. (pg 24, ln 19 – pg 25, ln 39 / pg 10, ln 17 – pg 11, ln 2)

8. The calibration documentation is quite thorough and comprehensive. I was prompted, however, to ask a few follow up questions regarding the variations in concentration dependence shown for dD. Is it possible one would get a different answer if concentration biases were adjusted first and VSMOW-scaling applied separately? It also appears that there are higher errors in dD at low isotopic values at all water vapor concentrations, not just at the low concentrations. Is it possible that lower precision at low isotope ratios causes the appearance of "irreproducibility" in the concentration dependence? Our calibration procedure does begin with concentration-dependence corrections (Sect. S2; note that the Appendix has been merged with the SI). We determined that VSMOW scaling of the concentration dependence-corrected delta values was not necessary (Figure S2.3). The Reviewer makes an interesting point about the apparent dD irreproducibility. However, we note that the "irreproducibility" at mid-range humidities (3000-8000 ppmv) only lie on one side of the correction curve (Fig. S2.2b), so it is an offset. On the other hand, variability on either side of the correction curve is observed for drier conditions (550-3000 ppmv; Fig. S2.2b), which indicates the low precision idea could be a possible explanation. We note that the uncertainty is only consequential for very low H2Ov mole fractions, where these depleted delta values are observed. We have added a sentence discussing the possible low precision and offset ideas (strike-through SI: pg 5, ln 5 / final SI: pg 5, ln 7).

Minor comments Page 1 Line 35 – no need for ":" after "include" The ":" has been removed.

Page 2 Line 10 –I had trouble distinguishing the conditions at a moisture source region from "surface H2Ov sources." Perhaps it might be more clear to say "an air parcel's moisture source region, including the geography of the source and its meteorological conditions?" This sentence has been changed to reflect the Reviewer's suggestion.

Page 2 Line 19 – I think "further exchange" is meant instead of "equilibrium?" "Equilibrium" has been replaced by "further exchange".

Page 3 Line 3 – I might remove "point in" before time, since I initially confused "point" with space. "Point in" has been removed.

Page 3 Line 6 – Perhaps best to say "higher…resolution" since aircraft is not as high resolution as slower-moving platforms. This is a good point, "high" has been changed to "higher".

Page 3 Line 11 – Perhaps best to say that "measurements of vertical profiles" were conducted. We have incorporated this suggestion.

Page 4 Line 28 – Perhaps "produce" for "emit?" Thank you for this suggestion, we have replaced "emit" with "produce".

Page 6 Figure 2 – Could the three analyzed flights be emphasized, perhaps by making the other flight lines dashed? This is a good suggestion, the case study flight paths are now indicated with solid lines, and all other flight paths are indicated with dashed lines.

Page 7 Table 1 – Table caption/title should provide some explanation of the "codes" and what is meant by "support study" Per the Reviewer's comment #5, we have modified the flights codes so that the flights are identified by their date.

Page 9 Line 13 – One of many examples of the great care that was taken in the analysis

Page 9 Line 31 – This appears to be the only place where "q" is used instead of "H2Ov." Consistency would help. Thank you for catching this. As noted in our response to major comment #4, we no longer include the Worden et al., 2007 equation that the Reviewer is referencing.

Page 10 Equations – This appears to be the only place "Rvap" is used instead of "Rv." Again, consistency would help. Thank you for catching this oversight. As noted in our response to major comment #4, we have removed the Stewart, 1975 equation that the Reviewer is referencing.

Page 11 Figure 4 – All the lines are "solid," thus it might be best to say the "black" line to distinguish it from the "pink" one. Also, I don't fully understand how the average mixing ratio is given by a gray envelope. Shouldn't the average be a point? Thank you for these suggestions. We have replaced "solid" with "black", and we have clarified that

the grey envelope indicates the inversion layer, which is defined by the average H2Ov mole fraction observed at zINV and zFT during the vertical profiles. (Fig 4 has been split into three figures for each of the case studies, Fig 7- CLR; Fig 9 – STC; Fig 12 – DBL)

Page 12 Line 7+ - Here is an example where it's easy for the reader to become lost in all the number-reporting. This paragraph would greatly benefit from a sentence that provides a bit more of a picture of what is going on physically. Thank you for this suggestion. We have added text to identify the important meteorological characteristics of each atmospheric layer on this day, and also decreased the amount of number-reporting. (pg 17, ln 6-19 / pg 7, ln 35 – pg 8, ln 9)

Page 12 Line 18 – I might say "predictions" instead of "theory." We have made this substitution.

Page 13 Figure 5 – Caption should explain what the shaded area for the inversion is and what the envelopes around the profiles are. The caption now indicates that the inversion layer (previously grey bands, now blue bands) lies between zFT and zINV, and that the shading around the isotope VP measurements correspond to total measurement uncertainty. (Now Fig 6)

Page 14 Line 6 – How was the average range of the inversion calculated? From how many days or which days of data? Similar to our changes to the caption of Fig. 4 (now Fig 7, 9 and 12) and Fig. 5 (now Fig. 6), we now specify that the bounds of the inversion layer range, which are unique to each flight, are defined by the average H2Ov mole fractions observed at zFT and zINV for each flight.

Page 14 Line 16 – I'm not sure I agree that the data start "tracking" the mixing line. There just aren't enough points for me to be convinced of that. Perhaps "approaches" the mixing line? We have made this substitution.

Page 14 Line 26 – This seems like an important argument explaining how is STC

different from Ray, and yet it is buried halfway down the page. Perhaps it could be moved up in the sub-section. We have moved this sentence to the second paragraph of Sect 4.2. (pg 21, ln 7 / pg 9, ln 18)

Page 15 Figure 6 – Most axes appear consistent across panels except for theta. Was this purposeful? Good catch! We have modified the theta range of Fig 6a (now Fig. 11a) for consistency.

Page 17 subtitle – I'm not really sure what "general observations" means. Would "observations from other flights" be more descriptive? We have renamed this subsection: "Airborne campaign observations of H2Ov isotopologues in the lower troposphere". (Now Sect. 3 title)

Page 17 Line 21 – which observations are used here for this argument? We now specify that the "DBL" case study observations are being discussed in this sentence.

Page 19 Line 19 – A sentence or couple words reminding the reader what "Scenario 1" is would be appreciated. We have revised the STC discussion paragraphs (see our response to major comment #2), and no longer reference cloud evaporation scenarios 1 and 2.

Page 19 Line 22 – I think "drier" is meant. Great catch, we did mean "drier". During our revision of this section, we have deleted "drier".

Page 21 Figure 9 – I would recommend dots (or some symbol) instead of vertical lines to indicate the values of dxs expected. Otherwise, it is not clear that the reader should look for the intersection of the various lines. We have revised Section 4.2 per major comment #2, and in doing so, we have removed Figure 9 from the manuscript.

Page 22 Line 24 – Excellent synthesis sentence: highlights the key point nicely.

Page 22 Line 36 – I disagree that these are some of the few data of this kind. There are quite a few studies that are not cited in this work. Please see my major comments for ideas. We have reworded this sentence to express that there are few studies that

report vertical profiles of d-excess.

Page 23 Line 24 – Kelsey et al 2018 also report dxs profiles. Thank you for this suggestion, we now reference the Kelsey et al, 2018 study here, and in the introduction.

Page 24 Line 4 – Perhaps "investigate" for "interrogate." We have made the suggested replacement.

Page 27 Line 5 – "To check calibration…" against? of? We have rephrased this sentence.

Page 30 Line 14 – The notation seems to change here. Should all isotopologues have subscript "v?" Thank you for catching this, we have added missing subscripted v's to the isotopologue ratios. (Now in the SI, Section S6)

Page 31 Figure D1 – At first I thought the black square was a symbol in the legend. It might be more clear simply to use the caption to say that the striped region shows the expected range for the observations, or something to that effect. Thank you for this observation. We have modified the legend to better represent the slanted line region, and we have added a sentence to the caption describing the saturation values that the slanted lines represent. (Now in the SI, Figure S6)

Please also note the supplement to this comment:
https://www.atmos-chem-phys-discuss.net/acp-2018-1313/acp-2018-1313-AC2-supplement.pdf

---

## Referee Report (RR1)

The revised manuscript is much more logical and easier to follow and the figures easier to interpret. There remain just a few places where I feel the wording is somewhat inaccurate, and I have flagged these below. Otherwise, my only remaining suggestion of significance is that the paper synthesize its main findings in a way that highlights the value of the deuterium excess measurements. Why should we as a community make d-excess measurements in addition to more traditional meteorological ones? What value is gained in having not just the individual isotope ratios but d-excess as well? The conclusions might, for example, touch upon the following points:

- D-excess appears to do a better job distinguishing mixing from Rayleigh processes in the FT on the CLR day.
- Only d-excess identifies the potential role of cloud in moistening the inversion layer during the STC day.
- In comparison, both d-excess and the individual isotope ratios are able to distinguish which air masses are mixing (interacting) on the DBL day.

P1, L27: It is not obvious to me what "moist processing" and "transport mixing" imply. Should this read "water phase changes, transport, and mixing?"

P2, L4: I suggest removing "dynamic, mesoscale" as climate feedbacks are usually regional/global.

P2, L29: Possible typo? "Variability in…co-vary?"

P3, L7: Satellite retrievals do provide HDO/H2O profiles outside the middle troposphere. The challenge with using them is the low spatial resolution in the vertical and lack of signal when clouds are present. Also, without H218O/H2O, there is no way to estimate d-excess.

P4, L33: Should LGR be the low-pass filtered timeseries if it has longer residence time (thus lower bandwidth) than Picarro?

Eqn 1: Should $H_2O_v$ in numerator also have a "Ray" subscript?

P 7, L18-19: Consider removing the last sentence. I'm not sure this is sufficiently substantiated.

P9, L32: It's not clear to me why mixing plots don't provide insight into the $Z_{FT}$ minimum. It seems they indicate mixing of FT and BL each with a distinct third end member (cloud influenced air?). Consider rephrasing/removing.

P10, L40: It is not clear to me that turbulent conditions are expected during formation of shallow stratocumulus. Some of the discussion in this section seems rather speculative.

P12, L25: Perhaps clarify that the mixing line connects with "lower altitude" delta values also observed during VP3.

Figs 3-4: Captions should note that isotope values from profiles are shown as a function of water vapor concentration.

Fig 6: Measurements in BL, INV, and FT are not indicated for reference but are the main point of the figure. This line can be deleted.

---

## Author Response (AR2)

Author responses are in blue text.

The revised manuscript is much more logical and easier to follow and the figures easier to interpret. There remain just a few places where I feel the wording is somewhat inaccurate, and I have flagged these below. Otherwise, my only remaining suggestion of significance is that the paper synthesize its main findings in a way that highlights the value of the deuterium excess measurements. Why should we as a community make d-excess measurements in addition to more traditional meteorological ones? What value is gained in having not just the individual isotope ratios but d-excess as well? The conclusions might, for example, touch upon the following points:

- D-excess appears to do a better job distinguishing mixing from Rayleigh processes in the FT of the CLR day.
- Only d-excess identifies the potential role of cloud in moistening the inversion layer during the STC day.
- In comparison, both d-excess and the individual isotope ratios are able to distinguish which air masses are mixing (interacting) on the DBL day.

We appreciate the reviewer pushing us to further clarify the value of these measurements to the atmospheric science community. This is an excellent opportunity and we've expanded the conclusions to highlight this value.

P1, L27: It is not obvious to me what "moist processing" and "transport mixing" imply. Should this read "water phase changes, transport, and mixing?"
We have made the suggested substitution.

P2, L4: I suggest removing "dynamic, mesoscale" as climate feedbacks are usually regional/global.
We have removed "dynamic, mesoscale".

P2, L29: Possible typo? "Variability in…co-vary?"
Great catch, we have modified this sentence.

P3, L7: Satellite retrievals do provide HDO/H2O profiles outside the middle troposphere. The challenge with using them is the low spatial resolution in the vertical and lack of signal when clouds are present. Also, without H218O/H2O, there is no way to estimate d-excess.
Thank you, we have modified this sentence to make some of these distinctions.

P4, L33: Should LGR be the low-pass filtered timeseries if it has longer residence time (thus lower bandwidth) than Picarro?
This sentence was misinterpreted. We did not apply a low-pass filter to the data. Rather, the longer residence time (slower flow and larger optical cavity) smooth the temporal signal in the LGR compared to the Picarro. We have rewritten it to improve clarity.

Eqn 1: Should $H_2O_v$ in numerator also have a "Ray" subscript?
We have added "Ray" as a subscript for clarification to $H_2O_v$.

P 7, L18-19: Consider removing the last sentence. I'm not sure this is sufficiently substantiated.
We have removed the indicated sentence.

P9, L32: It's not clear to me why mixing plots don't provide insight into the $Z_{FT}$ minimum. It seems they indicate mixing of FT and BL each with a distinct third end member (cloud influenced air?). Consider rephrasing/removing.

We have clarified this sentence so that it reads that the mixing lines do not identify the source responsible for the d-excess minimum at $z_{FT}$.

P10, L40: It is not clear to me that turbulent conditions are expected during formation of shallow stratocumulus. Some of the discussion in this section seems rather speculative.
We have reworded this section so that it is clear that a key characteristic of stratocumulus clouds is the cloud-top longwave cooling that maintains and enhances in-cloud turbulence (pg 10, ln 33-34).

P12, L25: Perhaps clarify that the mixing line connects with "lower altitude" delta values also observed during VP3.
Thank you for identifying this; we have clarified this sentence.

Figs 3-4: Captions should note that isotope values from profiles are shown as a function of water vapor concentration.
We have added that the delta and d-excess signatures are plotted versus $H_2O_v$ mole fraction.

Fig 6: Measurements in BL, INV, and FT are not indicated for reference but are the main point of the figure. This line can be deleted.
It should read the altitudes of BL, INV, and FT are indicated for reference. This is edited.